# Toward Training Superintelligent Software Agents through Self-Play SWE-RL

**Yuxiang Wei** [12]  **Zhiqing Sun** [3]  **Emily McMilin** [1]  **Jonas Gehring** [1]  **David Zhang** [1]  **Gabriel Synnaeve** [1]
**Daniel Fried** [14]  **Lingming Zhang** [2]  **Sida Wang** [1]

## Abstract

While current software agents powered by large language models (LLMs) and reinforcement learning (RL) can boost programmer productivity, their reliance on human-curated training data and environments creates a fundamental barrier to superintelligence. In this paper, we present Self-play SWE-RL (SSR), a first step toward training superintelligent software agents under minimal data assumptions. SSR requires only access to sandboxed repositories with source code and dependencies, no need for human-labeled issues or test commands. Grounded in real-world codebases, a single LLM agent is trained via RL in a self-play setting to inject and repair increasingly complex bugs. The bugs are formally specified by test suite improvements proposed by the agent rather than natural language issue descriptions. On the SWE-bench Verified and SWE-Bench Pro benchmarks, SSR achieves clear self-improvement (+10.4 and +7.8 points) and consistently outperforms the human-data baseline throughout training, generalizing to natural language bug descriptions not seen in training. Overall, our results point toward a paradigm where agents autonomously gather extensive learning experiences from real software repositories, ultimately enabling superintelligent systems that exceed human capabilities in understanding, modifying, and creating software from scratch.

## 1. Introduction

Software engineering agents (Yang et al., 2024; Wang et al., 2025b; Moatless Tools, 2024; Zhang et al., 2024; Xia et al., 2025a;b; Zhang et al., 2025; Luo et al., 2025; Yang et al., 2025b; FAIR CodeGen team et al., 2025; ByteDance Seed Team, 2025; Mistral AI, 2025) based on large language models (LLMs) have advanced rapidly and are boosting developer productivity in practice (Frunza, 2025). To improve LLMs' agentic ability, reinforcement learning (RL) with verifiable rewards has become the focal point. SWE-RL (Wei et al., 2025) is the first open RL method to improve LLMs on software engineering tasks using rule-based rewards and open software data. Since then, a variety of open LLMs focused on agentic RL have been released, including DeepSWE (Luo et al., 2025), DeepSeek V3.1 (DeepSeek AI, 2025), MiniMax M1/M2 (Chen et al., 2025a), Kimi K2 (Kimi Team et al., 2025), and Code World Model (FAIR CodeGen team et al., 2025). However, both the data and the environments used to train these agents, such as the issue descriptions and test cases, are heavily based on human knowledge or annotations. Even with RL applied, the resulting agents primarily learn to replay and refine human software development traces rather than independently discovering new classes of problems and solutions. Moreover, such curated training signals can be unreliable without extensive human inspection, as evidenced by the need for human-verified evaluation subsets like SWE-bench Verified (OpenAI, 2024). Consequently, this dependence on human knowledge and curation is unlikely to scale indefinitely, making it difficult for software agents to achieve the open-ended or superintelligent capabilities that purely self-improving systems might attain.

Although recent efforts such as SWE-smith (Yang et al., 2026a) and BugPilot (Sonwane et al., 2025) explore the use of LLMs for large-scale synthetic bug generation, these methods often hold stronger human-data assumptions, such as access to test suites and parsers, thus suffering the same aforementioned scalability limitations, and depend on teacher models for distillation. In addition, such existing methods typically rely on static bug generation pipelines without any consideration of the model being trained, limiting their ability to generate maximally informative examples as the agent improves. As a result, the system cannot continually self-improve.

In contrast, some of the most compelling examples of superintelligent AI arise from self-play. Following AlphaGo (Silver et al., 2016), AlphaZero (Silver et al., 2017) achieves

---

[1]Meta FAIR [2]University of Illinois Urbana-Champaign [3]Meta TBD Lab [4]Carnegie Mellon University. Correspondence to: Yuxiang Wei <ywei40@illinois.edu>.

*Proceedings of the 43rd International Conference on Machine Learning*, Seoul, South Korea. PMLR 306, 2026. Copyright 2026 by the author(s).

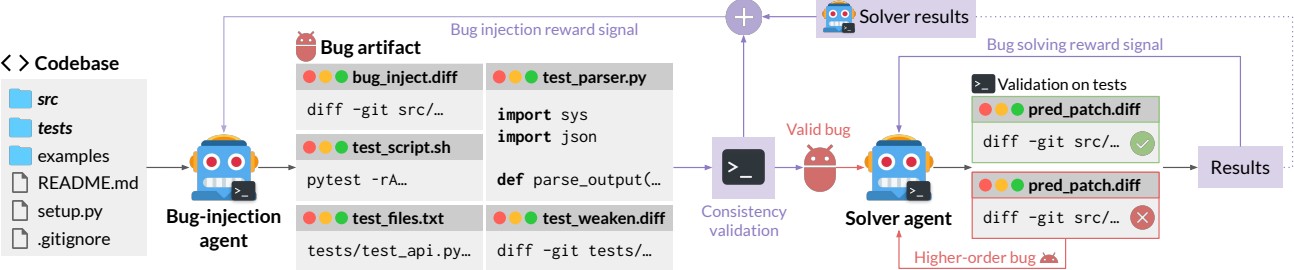

*Figure 1.* Overview of Self-play SWE-RL.

self-improvement in Go, chess, and shogi through self-play with only the game rules as input, showing that exploring and exploiting the implications of these game rules using RL can reach superhuman play. Recently, researchers have started to adopt self-play in open domains (Zhao et al., 2025; Huang et al., 2025; Kuba et al., 2025; Lin et al., 2025; Chen et al., 2025b; Liu et al., 2026; Haluptzok et al., 2023), some impressively but implausibly using nearly zero external data and relying on LLM introspection instead. Absolute Zero (Zhao et al., 2025) trains a single reasoning model to propose coding tasks that maximize its own learning progress and improves reasoning by solving them. Similarly, R-Zero (Huang et al., 2025) co-evolves a challenger and a solver to improve LLMs' reasoning on multiple domains. LSP (Kuba et al., 2025) also shows that pure self-play can enhance LLMs' instruction-following capability. However, such "zero" self-play cannot acquire knowledge beyond the fixed environment rules and the model's existing knowledge. Instead, SPICE (Liu et al., 2025) performs corpus-grounded self-play to interact with the external world for diverse feedback, which improves LLMs' general reasoning ability and outperforms ungrounded methods. For a thought experiment, consider what a human can learn by only interacting with the Python interpreter like in Absolute Zero. While they can learn all the intricacies of Python, they cannot learn the much greater knowledge and experiences contained in real-world codebases not inferable from Python semantics. This raises a natural question for software engineering: can we build software agents that, grounded in extensive real-world repositories, learn primarily from their own interaction with diverse code environments rather than from human-curated training data?

Inspired by these developments, we propose Self-play SWE-RL (SSR), a first step toward superintelligent software engineering agents that learn from their own experience grounded by raw codebases. SSR assumes only access to a corpus of sandboxed environments, each containing the source repository and its dependencies, without any knowledge about existing tests, test runners, issue descriptions, or language-specific infrastructure. In practice, each input to SSR consists solely of a pre-built Docker image. As shown in Figure 1, a single LLM policy is instantiated in

two roles by different prompting: a bug-injection agent and a bug-solving agent, both having access to the same set of tools adapted from Code World Model (CWM) (FAIR CodeGen team et al., 2025), including Bash and an editor. When the model plays the bug-injection role, it explores the repository, discovers how to run tests, and constructs a *bug artifact* that formally specifies a bug via a standard suite of artifacts: (1) a bug-inducing patch over code files, (2) a test script, (3) test files, (4) a test parser script, and (5) a test-weakening patch over test files. These artifacts are validated through a series of consistency checks and then handed to the solver role. Then the model plays the solver role, where it sees only the reversed test-weakening patch as a formal specification of the required behavior and must produce a repair patch that satisfies all specified tests. Both roles share parameters and are trained jointly with RL. We further introduce *higher-order bugs* constructed from the solver's own failed repair attempts. They enrich the training distribution by exposing the system to increasingly layered and realistic failure patterns that resemble the multi-step, interdependent edits required in real-world software development. Through comprehensive self-play, the model itself can generate an evolving curriculum of diverse, challenging bugs that naturally reflects its online and changing policy, which static bug-generation pipelines cannot provide.

We evaluate SSR on the widely-used SWE-bench Verified benchmark (Jimenez et al., 2024; OpenAI, 2024) and the more complex SWE-Bench Pro benchmark (Deng et al., 2025) using CWM-sft—the pre-RL checkpoint of CWM—as the base model. Our results indicate that SSR consistently surpasses the "human-data" baseline on both benchmarks over the entire training trajectory. This baseline is trained with the same hyperparameters as SSR and uses the same environment images, but it receives human-authored or human-curated issue descriptions together with the corresponding test suites and test commands for reward computation, following the setting of CWM 's agentic SWE-RL training. Furthermore, our results show that while self-play is effective across various bug-injection strategies, the choice of strategy introduces subtle but meaningful differences in training effectiveness. Vanilla bug-injection prompting causes the injected bugs to collapse into superficial one-line

modifications, yielding weak learning signals. Conversely, encouraging aggressive code removals and leveraging insights from historical changes improve the quality and effectiveness of learning. Finally, we demonstrate that self-play training provides superior performance compared with repair-only training, which conducts RL exclusively on bugs proposed by the proposer from earlier self-play iterations without engaging in new bug generation.

Overall, Self-play SWE-RL suggests a first step toward superintelligent software agents that autonomously accumulate vast learning experience from software repositories, eventually enabling systems that surpass human capabilities in understanding, improving, and generating software.

**Conflict of Interest Disclosure.** Several authors of this paper are employed by Meta, which leads the development of Code World Model (CWM) (FAIR CodeGen team et al., 2025) and CWM-sft (FAIR CodeGen Team, 2025), which are used as the base model and comparison points evaluated in this paper. The work also uses infrastructure and environment images associated with CWM.

## 2. Self-Play SWE-RL

### 2.1. Overview

The core idea of Self-play SWE-RL (SSR) is to allow LLM agents to self-improve through an iterative cycle of solving self-generated bugs and creating more complex challenges. As shown in Figure 1, the same LLM policy is divided into two roles: a bug-injection agent and a bug-solving agent. Both roles have access to the same containerized environment and set of tools, such as Bash and an editor, but are presented with different task specifications and requirements, where the implementation of the tool-using scaffold follows CWM (FAIR CodeGen team et al., 2025). In detail, the bug-injection agent receives a sandboxed environment of a raw codebase. Its task is to introduce a bug by producing an artifact containing the necessary files. The artifact's consistency—ensuring the bug is present and reproducible—is then validated through execution. A bug artifact that passes the consistency checking is considered valid and is presented to the solver agent, where the final solution patches are validated against the tests defined by the bug, with failure attempts treated as "higher-order" bugs for the agent to make another attempt on a different context. Eventually, the bug injection reward signal combines the consistency validation and solver results to incentivize better proposals, while the bug solving reward signal leverages the testing results. The same underlying policy LLM is jointly updated on both signals.

### 2.2. Input Assumptions

A key design principle of SSR is to minimize the required prior knowledge about the codebase, making the approach broadly applicable to diverse software projects. We only assume access to a corpus of Docker (Merkel, 2014) images containing source code repositories with dependencies installed. Notably, we do not assume access to test parsers, existing tests, commands to execute the test suite, or any prior knowledge about the programming language or test framework. The bug-injection agent is responsible for discovering how to run tests, creating test parsers, and understanding the test suite structure entirely through environmental interaction. This minimal assumption set ensures that SSR can be applied to arbitrary codebases with minimal setup overhead.

### 2.3. Agentic Bug Injection

**Bug artifact.** We specify a software bug for both training and evaluation purposes as an artifact of files that can validate the existence of a bug and a fix, parse the test results, define the bug changes, and hide the bug by weakening existing tests. Concretely, it involves the following files:

- **test_script.sh**: a bash script that runs the test suite to detect bugs and validate fixes.

- **test_files.txt**: a list of oracle test files that are always reset to their original version before running the test suite. After applying a model-generated patch, we run the test suite to verify correctness. Resetting the test files ensures evaluation integrity: even if an agent modifies or "games" the test files rather than properly fixing the bug, the original tests are restored for evaluation.

- **test_parser.py**: a Python script that parses test output and produces a detailed JSON mapping of each test ID to its result (passed / failed). The parser can be implemented in any programming language, irrelevant to the codebase's language; we use Python for its simplicity.

- **bug_inject.diff**: a git diff patch that introduces bugs into the existing codebase.

- **test_weaken.diff**: a git diff patch that removes or weakens tests to hide the bug from the test suite. This simulates realistic bugs that escape detection by existing tests and creates a test gap so that reversing it defines the expected behavior the fix must satisfy, serving as a specification for the solver. Figure 2 shows an example.

**Complex bug generation.** In SSR, bug generation is an agentic task where the agent interacts with the execution environment with tools to produce a bug artifact, further validated for consistency and provided to the solver agent. Based on the reward signal obtained from consistency checking and the feedback from the solver agent, the bug-injection

```
● ● ● test_weakening_patch.diff
@@ -957,6 +957,5 @@ class TestProphetWarmStart:
       daily_univariate_ts.iloc[:500], show_progress=False
   )
       m2 = Prophet(mcmc_samples=100, stan_backend=backend).fit(
-          daily_univariate_ts.iloc[:510], init=warm_start_params(m), show_progress=False
+          daily_univariate_ts.iloc[:510], show_progress=False
   )
-      assert m2.params["delta"].shape == (200, 25)
```

```
● ● ● oracle_test_patch.diff (weakening patch reversed: git diff buggy original -- {test_files})
@@ -957,5 +957,6 @@ class TestProphetWarmStart:
       daily_univariate_ts.iloc[:500], show_progress=False
   )
       m2 = Prophet(mcmc_samples=100, stan_backend=backend).fit(
-          daily_univariate_ts.iloc[:510], show_progress=False
+          daily_univariate_ts.iloc[:510], init=warm_start_params(m), show_progress=False
   )
+      assert m2.params["delta"].shape == (200, 25)
```

*Figure 2.* Example test weakening patch and its reversal as the oracle test specification for the solver.

agent learns to create higher-quality bugs that are more consistent and tailored for the solver through RL.

One crucial property for a good bug-injection agent is its ability to generate diverse bugs that capture the complexity of real-world software development, thereby training the solver agent across a comprehensive spectrum of software debugging and engineering scenarios. To achieve this, we adopt two simple strategies: instructing the agent to (1) **remove code files or hunks** from the codebase or (2) selectively **revert historical code changes** leveraging insights from git logs. Fixing these bugs requires adding or modifying code, which are among the most frequent change types in real-world software evolution. For both approaches, we require the agent to perform compatibility fixes to ensure the project remains runnable, so that any resulting bug reflects a true semantic error rather than a trivial syntax issue. With these bug injection methods, the solver agent is forced to understand how the repository is built by recovering the correct original code from a broken codebase. Figure 3 shows the example bug-injection patches and the key reasoning and action traces for the two methods.

Additionally, we introduce some control parameters, such as the minimum number of changed files, passing tests, and failing tests after injection to ensure the quality and complexity of the bug. See §D.1 for the detailed prompts.

**Consistency validation.** Each first-order bug artifact is validated against a set of execution rules to ensure it is meaningful. Figure 4 shows some key steps. Below, we describe the complete checks:

- **Test files existence and coverage:** all the test files must exist in the original repository and must be a superset of the files touched by the test weakening patch.

- **Test parser validity:** the test parser test_parser.py, used throughout subsequent validation steps, must reliably convert raw test output into a detailed JSON mapping from test names to their statuses (passed or failed).

- **Test script validity:** when executed on the original codebase, the test script test_script.sh must produce a list of passing tests, and the total number of the tests should exceed min_passing_tests. Test results are parsed using the test parser.

- **Bug scope:** the bug-injection patch bug_inject.diff must produce a minimum number of changed files, controlled by the parameter min_changed_files.

- **Bug validity:** at least min_failing_tests tests that pass in the original codebase must fail after applying bug_inject.diff.

- **Test weakening validity:** some tests that fail in the buggy state must pass after applying the test weakening patch test_weaken.diff.

- **Inverse mutation testing:** this check verifies that each file in the bug-injection patch is necessary to trigger the bug. We first collect the set of failing tests triggered by the complete bug. Then, for each modified file in isolation, we reset to the full buggy state, revert only that file to its fixed version, and run the non-weakened oracle tests. If reverting a file causes at least one failing test to pass, the file contributes to the bug; otherwise, the check fails. This approach inverts traditional mutation testing (DeMillo et al., 1978), which measures test suite quality by checking if it can reject random mutations

Bug artifacts passing the consistency checks are considered valid and are reformatted similarly to SWE-bench (Jimenez et al., 2024) instances, including pass-to-pass and fail-to-pass test specifications.

**Higher-order bugs.** Generating complex bugs via removal, while flexible, can be restricted in incentivizing the solver agent to learn different editing patterns, as the solver basically needs to reconstruct code files or hunks from scratch. It is also less natural compared to real-world bugs in public repositories (such as those from GitHub PRs) where the original code typically does not consist of unimplemented parts. Meanwhile, the generated bugs can be very complex to solve in the initial attempts as a result of removing large portion of code.

To make sure our bug-injection approach is scalable, in the sense that running the agent for extended periods should consistently yield new, distinct bugs rather than exhausting existing patterns, we further incorporate **higher-order bugs** in training the solver agent, where we collect failed initial bug-solving attempts from the solver agent and form new bug artifacts using the new buggy states derived from the

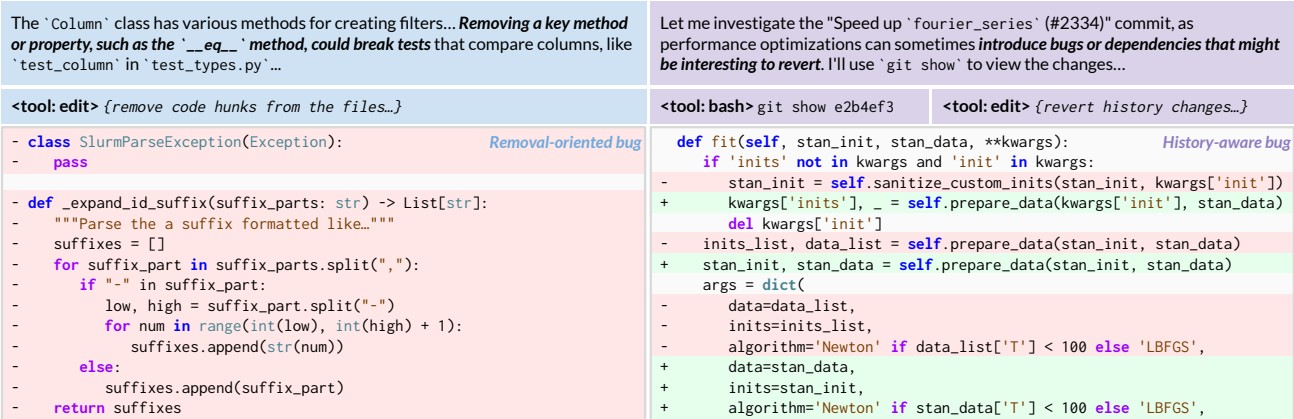

*Figure 3.* Bug-injection patches generated by code hunk removal (left) and historical change reversion (right).

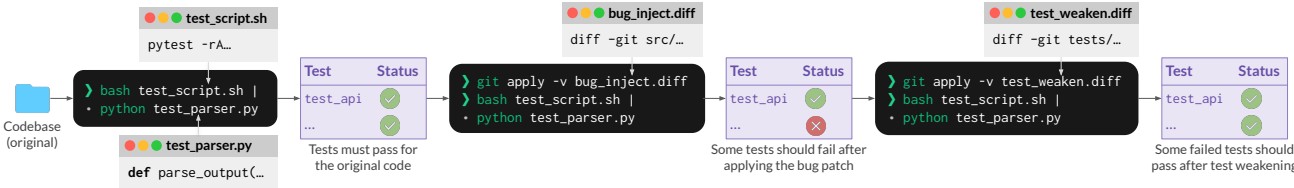

*Figure 4.* Key consistency checks applied to validate bug artifacts, the full set described in the text.

solver attempts. These higher-order bugs mimic how developers unintentionally write buggy code in their natural development process, enabling the agent to generate bugs across all aspects of its own coding capabilities.

**Reward design.** The bug-injection agent receives rewards based on the quality and difficulty of generated bugs, which we measure through consistency validation and solver performance. Let $s \in [0, 1]$ denote the *solve rate*, defined as the fraction of solver attempts that successfully fix the bug entirely. The bug-injection reward $r_{\text{inject}}$ is defined as:

$$r_{\text{inject}} = \begin{cases} -1.0 & \text{if consistency validation fails,} \\ -\alpha & \text{if } s = 0 \text{ or } s = 1, \\ 1 - (1 + \alpha)s & \text{if } 0 < s < 1 \text{ (ideal difficulty),} \end{cases}$$
$$(1)$$

where $\alpha \in (0, 1)$ is a hyperparameter controlling the penalty magnitude for valid bugs with degenerate solve rates ($s = 0$ or $s = 1$), set to $0.8$ in our experiments.

This reward function is adapted from (Zhao et al., 2025), incentivizing the agent to create consistent bugs that are neither trivially solvable ($s = 1$) nor impossibly hard ($s = 0$), with maximum reward achieved when bugs are challenging yet solvable with low but non-zero success rates. Notably, we penalize valid bugs with extreme solve rates by $-\alpha$ instead of $-1.0$, ensuring the agent still receives informative gradients from these cases and can learn to better calibrate bug difficulty over time. See §A.1 for the optimal target solve rate and §A.2 for some theoretical strategies that the

bug-injection agent can adopt to maximize its reward.

### 2.4. Agentic Bug Repair

**Construct the buggy codebase.** If a bug artifact passes the consistency validation, the solver agent will start the bug repair process by interacting with the buggy codebase to produce a bug-fixing prediction patch. Figure 5 illustrates the pipeline for constructing such buggy codebase.

Starting from the original codebase, we first apply `bug_inject.diff` to introduce the bug. Next, we apply `test_weaken.diff` to hide the bug from the existing test suite. For higher-order bugs, we additionally apply the failed prediction patch `pred_patch.diff` from a prior solver attempt. Finally, to prevent information leakage through git history, which could otherwise lead to hacking (jacobkahn, 2025), we reinitialize the repository by removing the `.git` directory and creating a fresh commit. The resulting buggy codebase forms the solver's starting environment.

**Initial prompt.** In addition to the buggy environment, the agent also needs an initial prompt to understand the task requirements. We simply provide the agent with a reversed version of the test weakening patch, with an example showed in Figure 2, telling it to satisfy the test requirements while making sure all relevant tests in the codebase can pass without errors. The detailed prompt template is shown in §D.2. Notably, we **do not synthesize issue descriptions in natural language forms**, as it is difficult to automati-

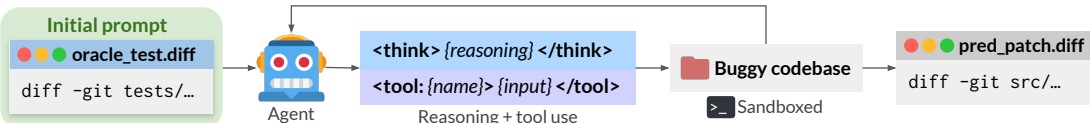

*Figure 5.* Constructing the buggy codebase from a valid bug artifact.

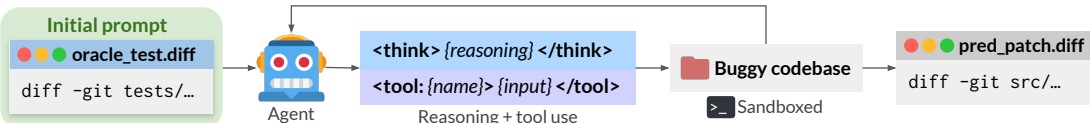

*Figure 6.* Agentic bug-repair process.

cally evaluate the quality of free-form natural language text. Furthermore, high-quality natural language issues are inherently scarce compared to the vast synthetic tasks that can be created by bug injection. As a result, the downstream improvements in issue solving, which operates on natural language issues, purely stem from learning to write test-passing code rather than simple in-domain generalization.

**Bug-repair process.** With the buggy codebase and the initial prompt, the bug-repair process is shown in Figure 6. Starting with the initial prompt, the agent interacts with the sandboxed buggy codebase by performing reasoning and using tools, until it produces a final prediction patch.

For each prediction patch, we evaluate its correctness using the pipeline demonstrated in Figure 7. Beginning with the original codebase, we first tag the current state as ssr-original to preserve a reference to the ground-truth repository. We then apply bug_inject.diff followed by test_weaken.diff to construct the buggy codebase. Next, we apply the predicted patch to the buggy codebase and restore the test files listed in test_files.txt from the ssr-original tag. This restoration step ensures evaluation integrity. Finally, we execute test_script.sh and pipe its output through test_parser.py to obtain a structured mapping of test results in JSON, determining whether the patch successfully resolves the bug.

As discussed in §2.3, some solver attempts may fail. These failed attempts are converted into **higher-order bugs** based on the new bug states derived by the agent. Higher-order bugs are provided to the solver agent to make additional attempts. We do not go beyond the second order to prevent a higher likelihood of overlapping with existing bugs.

**Reward design.** For the bug-solving agent, we employ a simple binary reward based on whether all the tests pass after applying the prediction patch:

$$r_{\text{solve}} = \begin{cases} +1 & \text{if all tests pass,} \\ -1 & \text{otherwise.} \end{cases} \quad (2)$$

## 3. Evaluation

### 3.1. Experimental Setup

**Base model.** We use Code World Model (CWM), a state-of-the-art open-weight 32B code LLM for agentic software engineering, as our base model. Specifically, we utilize the CWM-sft (FAIR CodeGen Team, 2025) checkpoint, which is obtained prior to its joint RL stage. This allows us to build on a checkpoint that has not been influenced by RL and fairly evaluate different learning strategies. We use the same LLM for bug generation and repair, where the shared policy weights are updated continuously during RL.

**Evaluation setup.** We conduct evaluation on SWE-bench Verified (OpenAI, 2024), a subset of SWE-bench (Jimenez et al., 2024) with 500 human-verified real-world software issues, and SWE-Bench Pro (public split) (Deng et al., 2025), which contains 731 publicly available software problems, aimed to capture realistic, complex, and enterprise-level tasks. We perform one attempt for each problem without parallel test-time scaling or ranking. We use temperature = 1.0 and top-p = 0.95 for evaluation.

### 3.2. Baseline Comparison

We compare the base model, baseline RL, and SSR on the SWE-bench Verified and SWE-Bench Pro benchmarks. Both baseline RL and SSR train on an identical set of environment images. The fundamental difference lies in what information each approach can access. Baseline RL, like standard agentic RL in CWM (FAIR CodeGen team et al., 2025), has access to natural language issue descriptions, pass-to-pass and fail-to-pass tests, and evaluation scripts, so the RL process simply checks whether solutions pass the given tests. In contrast, SSR operates with only bare environment images, requiring the model to discover problems, formulate solutions, and validate them entirely through self-play without any provided descriptions or tests.

Figure 8 presents the results. We first observe that SSR achieves steady self-improvement throughout training, despite having no access to task-specific training data. This

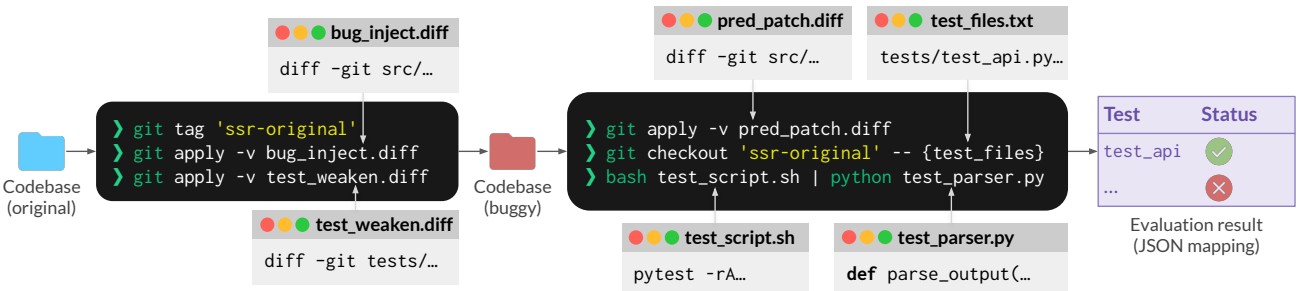

*Figure 7.* Evaluating the correctness of model-predicted patches.

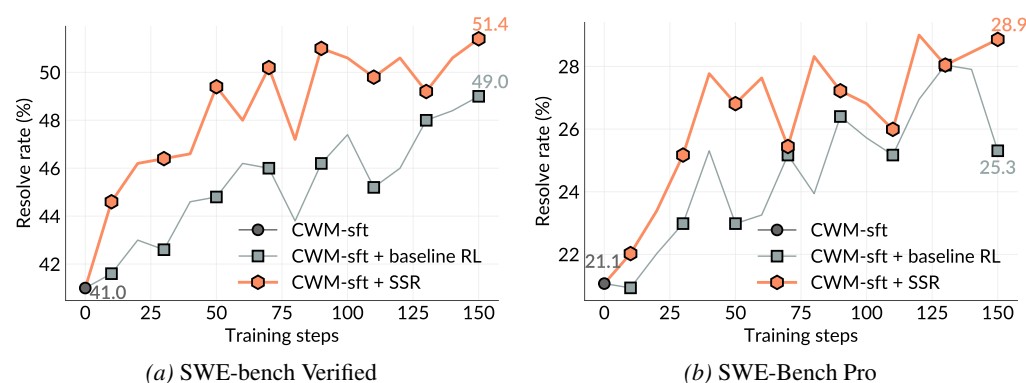

*(a)* SWE-bench Verified

*(b)* SWE-Bench Pro

*Figure 8.* Baseline comparison over the course of training.

demonstrates that LLMs can self-improve their software engineering ability, such as issue-solving, purely through interaction with raw codebases. Furthermore, SSR consistently outperforms baseline RL on both benchmarks across the entire training trajectory, indicating that self-generated tasks provide richer and more effective learning signals than human-engineered data.

### 3.3. Ablation of Self-Play Components

To isolate the contribution of self-play, we compare the SSR with an injection-only and a repair-only variant, both trained using standard RL. In the injection-only setting, the agent is only trained for bug generation, still using the SSR reward but with no training signal from repair trajectories; for repair-only, the agent is exclusively trained to solve bugs, where the bug data comes from the valid, solvable, and deduplicated bugs from prior self-play runs.

Figure 9a shows that self-play is the best. Injection-only training degrades the performance since it does not learn from any bug-solving attempts. Repair-only training is also inferior because it lacks the evolving task distribution produced by self-play. In contrast, self-play requires the agent not only to repair bugs but also to propose challenging ones, which itself embodies substantial learning: identifying passing tests, breaking functionality in meaningful ways, and weakening tests to hide the bug. These activities continually

expand the training signal and expose the model to new failure modes. These results indicate that an evolving and online process of bug generation and bug solving is essential for sustained improvement.

### 3.4. Ablation of Bug Injection Methods

We also study how different bug-injection strategies affect downstream performance. Figure 9b compares three variants (full prompts in §D.1):

- **Direct-injection**: naive prompting to introduce bugs without detailed guidance.

- **Removal-only**: instructing the agent to remove substantial code hunks or files while ensuring the project remains runnable.

- **Removal + history**: randomly sampling between two prompts, one using the removal-only strategy and the other using a history-aware strategy that reverts selected historical changes from git logs.

From the figure, SSR proves effective across all injection strategies, demonstrating its generalizability. However, the effectiveness of different methods varies in subtle but meaningful ways. First, direct-injection leads to the worst result because it tends to produce trivial bugs with superficial one-line modifications (e.g., var = 0 → var = 1) that offer little

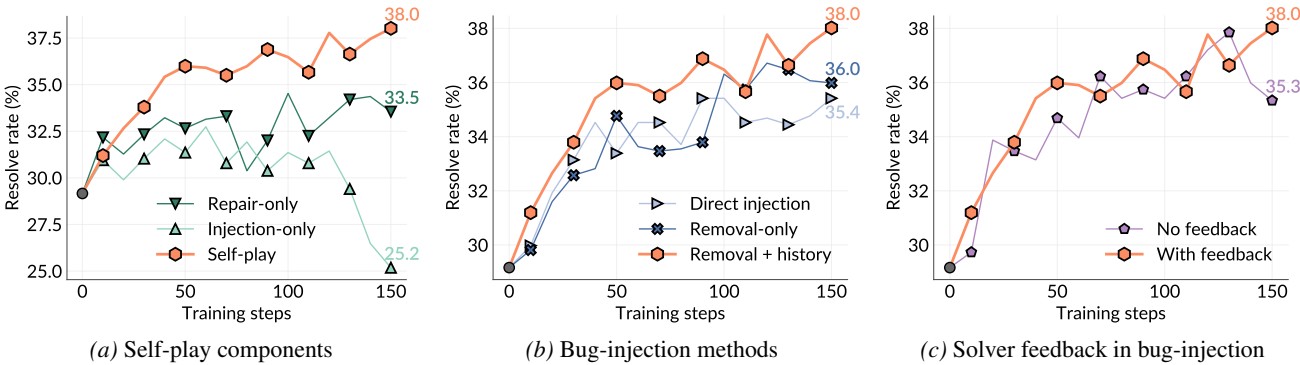

*(a)* Self-play components      *(b)* Bug-injection methods      *(c)* Solver feedback in bug-injection

*Figure 9.* Ablation studies. The resolve rate is averaged over 1231 tasks from 500 SWE-bench Verified and 731 SWE-Bench Pro tasks.

training signal. By contrast, the removal-only strategy generates stronger bugs, forcing the solver to reconstruct missing functionality and thereby deepening its understanding of repository structure. Finally, "removal + history" achieves the best overall performance by leveraging more historical changes that incorporate more realistic and diverse bug patterns. These findings highlight the importance of carefully designing bug-injection mechanisms for a more challenging and instructive curriculum.

### 3.5. Reward Ablation

As discussed in §2.3, the bug-injection reward incorporates feedback from the solver role based on the bug's solve rate. In this ablation, we compare against a binary reward variant that ignores this feedback, assigning a value from $\{-1, 1\}$ based solely on whether the generated bug passes consistency checks. As shown in Figure 9c, solver feedback provides only a slight and largely negligible advantage over the consistency baseline. We hypothesize that this is because solve-rate feedback is weak and noisy. For the bug-injector, it is difficult to learn the factors affecting the solve rate from a single number to generate the ideal bugs. Figure 10 shows that the expected reward estimated using $G$ noisy samples is smoothed compared to Equation (1), thus a larger range of solve rates has similar expected rewards. As for the solver, real data also has various solve rates, where too-easy or too-hard problems reduce efficiency but are not harmful.

Notably, even without solver feedback in the reward, the bug-injection agent still uses the evolving policy continuously updated from both bug-generation and bug-solving. This online joint-learning enables it to generate an evolving curriculum that naturally reflects the agent's current policy, as discussed in §3.3, which static bug-generation pipelines cannot provide.

## 4. Related Work

**Agent scaffolding for software engineering.** Since the release of SWE-bench (Jimenez et al., 2024), which frames real GitHub issues as end-to-end repository-level repair tasks, researchers have designed various scaffolds on improving LLMs' issue-solving ability. Two families of scaffolds have emerged. Agentic scaffolds (Yang et al., 2024; Wang et al., 2025b) involves an LLM driving the core decision-making process based on its tool-mediated interaction with the sandboxed environment. A canonical example is SWE-agent (Yang et al., 2024), which introduces the concept of agent–computer interface, an abstraction layer between LLMs and computers, to enhance LLM agents' ability inside computer environments. By contrast, pipeline-based scaffolds (Xia et al., 2025a; Moatless Tools, 2024) decompose the task into human-defined stages, which typically involves fault localization, patch generation, and patch selection. These workflows trade generality for stability and cost control. Due to the huge design space of agentic scaffolds, recently, researchers have proposed self-improving agents (Zhang et al., 2025; Wang et al., 2025a) to improve their own scaffolds via iterative offline learning, along with LIVE-SWE-AGENT (Xia et al., 2025b) that self-evolves its scaffold on the fly through tool creation.

**Training software agents via reinforcement learning.** Recent work improves the core model competence for software engineering tasks through reinforcement learning (RL). SWE-RL (Wei et al., 2025) applies RL to "software evolution" data with lightweight verifiable rewards, yielding a model that improves solve rates on software engineering and reasoning tasks. DeepSWE (Luo et al., 2025) trains a 32B open-weight agent from scratch using pure RL on containerized software environments with execution-based reward. Code World Model (CWM) (FAIR CodeGen team et al., 2025) is trained as a reasoning agent whose trajectories interleave reasoning and tool use, achieving state-of-the-art results among 32B models with test-time scaling. Increasingly, newly released open LLMs emphasize agentic intelligence

and are trained extensively on software engineering data and environments, including DeepSeek V3.1 (DeepSeek AI, 2025), DeepSeek V3.2 (DeepSeek-AI et al., 2025), Mini-Max M1/M2 (Chen et al., 2025a), Qwen3-Coder (Yang et al., 2025a), Kimi K2 (Kimi Team et al., 2025), SWE-1.5 (Cognition, 2025), GLM-4.5 (Zeng et al., 2025), Devstral (Mistral AI, 2025), and MiMo-V2-Flash (Xiao et al., 2026).

**Data and environments for training software agents.** Robust software agents need a large corpus of diverse executable software environments and task descriptions for training. Recent work has converged on two complementary lines: real data collection and synthetic data generation. SelfAPR (Ye et al., 2023) is an early example of project-specific synthetic repair-data generation. It perturbs previous versions of the program under repair to create training samples and incorporates test execution diagnostics into the model input, enabling the repair model to capture both project-specific knowledge and fault-type information. SWE-Gym (Pan et al., 2025) is a manually curated software issue dataset with 2.4k real Python tasks from 11 projects. Each instance ships an executable environment and unit tests, enabling both agent training and verifier training for inference-time best-of-$n$ selection. R2E-Gym (Jain et al., 2025) scales beyond human-curated issues by back-translating commits into over 8k executable tasks with auto-generated tests and problem statements. SWE-rebench (Badertdinov et al., 2026) automates continuous extraction of fresh interactive tasks from GitHub (21k+ Python tasks) and maintains a rolling, decontaminated evaluation window with a public leaderboard. SWE-smith (Yang et al., 2026a) automatically converts real Python repositories into a training task through environment creation, bug synthesis, and issue generation, providing lightweight runnable environments and a 50k task corpus across 128 repos. Finally, BugPilot (Sonwane et al., 2025) targets difficult synthetic bugs by having agents implement features that unintentionally break tests, producing more natural failures that improve training efficiency.

## 5. Conclusion

We present Self-play SWE-RL (SSR), a first step toward training superintelligent software agents through self-play. Operating on minimal data assumptions by requiring only sandboxed environments with source code and dependencies, SSR enables agents to autonomously generate and solve increasingly complex bugs without relying on human-curated issue descriptions or test suites. Our evaluation on SWE-bench Verified and SWE-Bench Pro benchmarks demonstrates that self-play enables steady self-improvement during training and outperforms human-data baselines over the course of training. Our results, albeit early, suggest a path where agents autonomously gather extensive learn-ing experiences from real-world software repositories, ultimately enabling superintelligent systems that exceed human capabilities in understanding how systems are constructed, solving novel challenges, and autonomously creating new software from scratch.

## Acknowledgement

We thank Taco Cohen, Ori Yoran, Zijian Wang, John Yang, Chen Zhu, Quentin Carbonneaux, Kunhao Zheng, Jannik Kossen, Dmitrii Pedchenko, Sten Sootla, Jordi Armengol Estape, and Zacharias Fisches for their valuable discussions and infra assistance; Don Landrum and Eslam Elnikety for their timely support in resolving container infrastructure issues; Josh Song and Elisa Cascardi for their prompt feedback in the paper review process.

We also appreciate the valuable feedback from the reviewers that help us improve the paper. This work was partially supported by NSF grant CCF-2131943.

## Impact Statement

This work studies self-play reinforcement learning as a way to improve software engineering agents without relying on human-curated issue descriptions or test suites. By enabling agents to autonomously generate and solve bugs in real-world codebases, this approach may reduce data collection costs and improve scalability of training for software engineering tasks. It could support more effective automated debugging and maintenance, particularly in settings where labeled data is scarce or outdated.

However, the ability to intentionally generate bugs and weaken tests also introduces risks, such as misuse for producing vulnerable or misleading code artifacts. Careful sandboxing, responsible deployment, and adherence to ethical guidelines are therefore necessary to ensure this technique is used safely and constructively.

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

# A. Analysis of the Challenger-Solver Game

For two-player zero-sum games like go and chess, extensive self-play can be expected to sufficiently explore and exploit the game rules to reach superhuman intelligence given enough compute and model capacity. For the challenger-solver game, we show that a sufficiently intelligent challenger has several dominant strategies so the solver learns nothing useful and the self-play stops progressing. We first show that the goal of the challenger is to set an optimal target solve rate, then we outline two types of dominant strategies of the challenger. The first is a truly dominant challenger but requires a powerful action space such as ours. The second is a tunnel-vision challenger that applies to other self-play settings as well such as Absolute Zero. Finally, we discuss the role of natural language and some practical mitigations that address these problems by strong grounding and limited self-play.

## A.1. The Challenger Has an Optimal Target Solve Rate

Reward (1) increases toward 0, but is negative at exactly 0. While this seems to encourage the challenger to target an arbitrarily small solve rate, "arbitrarily small" cannot be distinguished from 0 by a small group of $G = 8$ samples so some margin is needed to achieve the optimum expected reward. We show that challenger reward (1) and $r(s) = s^a(1-s)^b$ both have a unique optimal target solve rate $p^* \in (0, 1)$ where $s$ is estimated by a group of $G$ trajectories, each with a target solve rate of $p$. To pick $p$ that maximizes the expected reward, the RL algorithm draws $k \sim \text{Binom}(G, p)$ samples to get the group's empirical solve rate $s = k/G$ and the challenger gets a reward $r(s)$. So the real goal of the challenger is to maximize the expected reward

$$R(p) = \text{E}_{k \sim \text{Binom}(G,p)}[r(k/G)] = \sum_{k=0}^{G} \binom{G}{k} p^k (1-p)^{G-k} r(k/G),$$

which is a degree $G$ polynomial in the target solve rate $p$. If $r(s) = s^a(1-s)^b$, then the optimal success rate is around $p^* \approx a/(a+b)$ with increasing precision for large $G$ as the binomial becomes more concentrated. If the reward says to avoid 0 or 1 more strongly like (1), then there will be an appropriate amount of conservatism to move $p$ towards the middle, especially with small $G$. While (1) appears different from the Beta reward $r(s) = s^a(1-s)^b$, they behave similarly when viewed in terms of the expected reward as a function of the target solve rate $p$ where the discontinuity of (1) disappears in expected reward. Figure 10 shows that for the expected reward, (1) is similar to the Beta reward with appropriate $a$ and $b$, and there is a unique maximum target solve probability $p^* \in (0, 1)$ around 0.2 for the challenger to achieve its optimal expected reward.

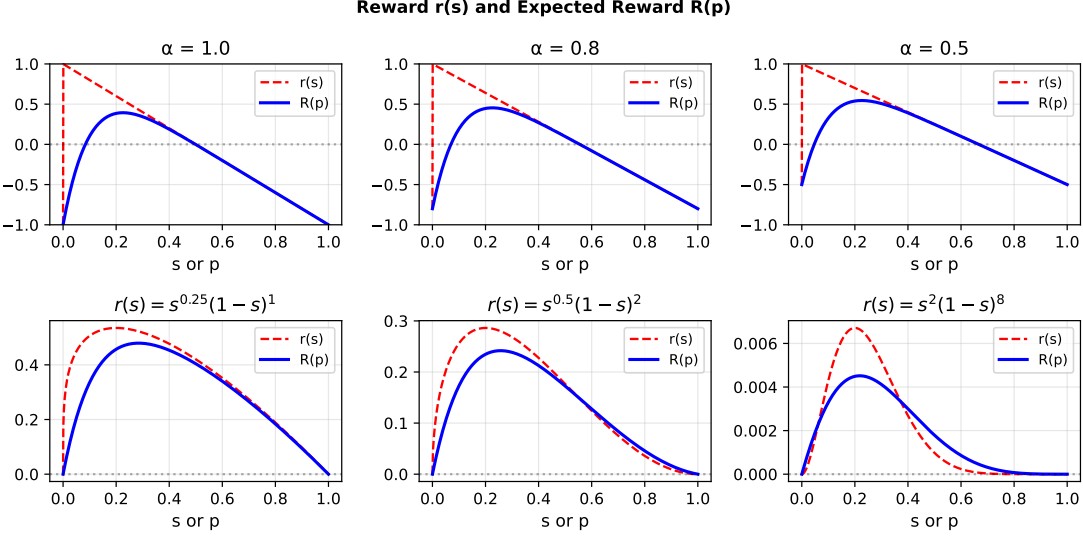

*Figure 10.* Expected reward $R(p)$ using $G = 8$ vs. the target solve rate $p$ plotted with the original reward function $r(s)$ for context. The expected reward applies a smoothing function and lessens the difference between various reward choices $r(s)$. While $\alpha$ only shifts the reward function, it can affect the relative weighting of other failure types (such as consistency or system error) so it is an appropriate single-purpose parameter.

### A.2. The Challenger Strategy

While the solver achieves its sensible optimal reward by solving all the problems, the challenger can employ several strategies to achieve its own optimal reward at the cost of the solver.

**The dominant challenger.** The challenger has a *dominant strategy* if it can output a question that is solved with probability $p^*$ and achieve its own optimal reward regardless of what the solver does. This is possible when the challenger has sufficient freedom to set the problem.

The most direct way is when the challenger is allowed to modify the tests, then one such test is **fail-randomly**: `if rand() < p: pass(); else: fail()`. If we try to fix this by requiring determinism, we can use the code text of the solution to generate a pseudo-random number instead,

$$\text{if rand(seed=hash(current\_code)) < p: pass(); else: fail()}$$

where the pseudo-random function returns the same number on the same code, which works as long as the sampled solutions are different. If we require that the original code must pass, the test can have a case for the original code too `if hash(current_code) = hash(original_code): pass()`, where the original code hash is a constant, and the current code is read from the files.

Even when the test is fixed, one strategy is the following:

1. Add the **fail-randomly** method (or its pseudo-random extension) to return the wrong output at random or just raise an exception.

2. Obfuscate the whole codebase so it is impossible for the solver to find and fix the change.

Here the solver cannot improve on the proposed problem assuming that we have good enough obfuscation so the solver cannot understand the code. The solver can make it worse but has no incentive to, so the best response for the solver is to do nothing[1]. It seems hard to prevent this theoretical behavior as long as the challenger has a Turing complete action space to take actions such as these. In practice, if the challenger does not have an awareness of its own setting, it may not be able to design such optimal strategies even if it is superintelligent.

**The tunnel-vision challenger.** Even if the challenger has a limited actions space, the challenger can still focus its tunnel-vision on one specific parameter to control the difficulty and does not cover the diversity that we want the solver to learn. Since the solver cannot keep improving on any specific task, learning eventually stops and the challenger achieves its optimal expected reward. For example, suppose that long multiplication is challenging for the solver, the challenger can just adjust the length of the operands until it is at the desired difficulty. Even as the solver learns, it presumably cannot learn to do infinity long multiplications and thus will be stuck at a desired accuracy as the length increases and the challenger is optimal. In the software setting, the challenger can hide a particular type of bug with increasing level of obfuscation. Another method is to chain a number of medium solve rate challenges together, and the solver has to solve all of them to pass, and the challenger can adjust the chain length appropriately without covering diverse challenges outside of this arbitrary and narrow tunnel of problems.

**Improving natural language communication with humans.** We argue that deep self-play without human language grounding or human interaction is unpromising, at least if the goal is to communicate well with current humans. Diplomacy (Meta Fundamental AI Research Diplomacy Team (FAIR) et al., 2022) argues that when learning to cooperate with humans, *self-play without human data is no longer guaranteed to find a policy that performs well with humans.* This is clearly demonstrated by the Ultimatum game, where the optimal play of always accept any positive offer is quite different from human behavior, which has fairness and ego built-in. Learning to solve human problems and fixing human issues where natural language plays a major role is an example of cooperating with humans. In any particular game where communication is needed, there are many sufficient *languages*, where many are more efficient than natural language because they can specialize to this setting only. While deep self-play can improve communication skills, it would not plausibly stay with and improve the quality of its human natural language even when initialized by a LLM trained on natural language. For some examples, (Kottur et al., 2017) shows effective communication in games needs not to be like human language or even compositional. In learning language games (Wang et al., 2016), humans are observed to adapt to a particular grounding and

---

[1] Thus not technically a dominant strategy, but still strong.

stop using standard language. To draw another analogy, we note that human languages also evolve and change to become unrecognizable over a long period of time,[2] or even over a short amount of time in the case of software engineering writing laden with Three-Letter Acronyms (TLA) evolved from self-play by closed teams incomprehensible to anyone outside.

**Mitigations.** Since the challenger has a dominant strategy, we do not desire enough self-play to fully explore and exploit the implications of the game rules, unlike for 2-player zero-sum games. However, a shallower challenger can still be beneficial by taking advantage of the model's ability to generate and explore diverse challenges and without learning to adopt its best strategy. In particular,

1. The challenger should be grounded in large and diverse real world data, such as all code repositories or documents. The goal is to skillfully pose natural and diverse challenges grounded in and inspired by real data.

2. Do not let the challenger diverge much from the initial instruction-based strategy and ensure that the self-play does not reach the tunnel-vision strategy. The challenger can still learn skills like passing the consistency checks and some calibration of question difficulty.

3. Do not attempt to improve on natural language skills using self-play without continued grounding on current human natural language. Gaining technical skills and code world model knowledge from self-play is desirable and plausible.

## B. Opposing Incentives in the Reward Design

The solve rate $s$ in Equation (1) is computed over all solver attempts (e.g., 8 attempts per bug). Each individual solver attempt receives the binary reward $r_{\text{solve}} \in \{+1, -1\}$ based solely on whether it succeeds. Given a bug with solve rate $s$, the expected solver reward is $\mathbb{E}[r_{\text{solve}}] = s - (1 - s) = 2s - 1$. For valid bugs with ideal difficulty ($0 < s < 1$), the injection reward simplifies to $r_{\text{inject}} = 1 - (1 + \alpha)s$. Consequently, the solver benefits from higher $s$ (easier bugs increase expected reward), while the injector benefits from lower $s$ (harder bugs increase injection reward), though bugs that are too hard ($s = 0$) are penalized to prevent unsolvable proposals. This adversarial pressure pushes the injector to propose bugs near the frontier of the solver's current capability. Note the optimal target difficulty depends on the sample size, explained in §A.

## C. Experimental Details

**Training details.** We implement SSR on top of the async CWM-RL infrastructure (FAIR CodeGen team et al., 2025) and adopt the same RL algorithm, with some hyperparameter modifications. We train the models on NVIDIA H100 SXM 80G GPUs, with a standard configuration of 512 GPUs for a single training run, with 64 GPUs for training and 448 GPUs for rollouts. Inspired by ScaleRL (Devvrit et al., 2026) and MiniRL (Zheng et al., 2025), we incorporate a "large-batch" and "small-policy-staleness" hyperparameter setup. By default, we employ a 131,072 maximum context window size for generation, pack training sequences by a maximum of 131,072 tokens, use a global batch size of 16M tokens, achieved through 16 gradient accumulation steps per optimizer step, and a typical group size of 8 for acceptable rollout latency, discard rollouts with more than 8 off-policy steps, and train the models (including baseline RL, SSR, and all ablation runs) for 150 global steps, roughly 2.5B tokens, with 30 steps of learning rate warmup toward $3 \times 10^{-6}$.

Our implementation employs the same tool-based scaffold as CWM with minor prompt adjustments, incorporating Bash and a search-replace editor. All environment images used in our experiments are identical to those in CWM.

**Evaluation noise.** While our method demonstrates consistent self-improvements across the training steps, we acknowledge the presence of evaluation noise, typically around 2% paired standard error on SWE-bench Verified. Readers can refer to Eval Arena (Wang et al., 2024) for a detailed discussion.

---

[2]A popular introduction is *The Power of Babel: A Natural History of Language*

# D. Prompt Templates

## D.1. Bug Injection

---

**Removal-oriented bug-injection**

You are working with a random commit from a code repository. Your goal is to introduce **complex bugs** into the codebase by removing multiple code files or
↪ code chunks and then removing tests to hide the bugs. The bugs will serve as training data for a bug-fixing AI system.

### Steps to follow

1. Understand the codebase, its functionalities, and the test suite / framework / command structure.
2. **Identify interesting test files**: find a set of test files that cover significant functionality in the codebase, making sure they involve at least
↪ {min_passing_tests} tests and test over {min_changed_files} code files.
3. **Set up the test command**: Create a test command that runs your selected tests. Make sure your test command can output the detailed test results,
↪ including which tests passed or failed (e.g., `pytest -rA`); ensure that the test execution takes less than 90 seconds. Dump the test command in a bash
↪ script (e.g., `test_script.sh`, which may involve additional setups like environment variables) for later use.
4. Trigger the test command, **directing the output to a log file** (e.g., using `bash test_script.sh > test_output_existing_tests.log 2>&1`) because the
↪ test output can be very long. View the log file (e.g., using `head` or `tail`) to verify the results.
5. If not all tests pass, it may be due to flakiness or environment issues; just exclude them from the test command or find another set until all selected
↪ tests pass (still needs to satisfy the minimum passing tests requirement: >= {min_passing_tests}).
6. Write a generic parser script in Python that can parse any test log file output by your test command. The test script should read from stdin the test log
↪ output and writes to stdout a JSON summary mapping each executed test case to either "passed" or "failed". Verify that your parser works. Example usage
↪ and output:

```bash
$ cat test_output_existing_tests.log | python3 parse_test_output.py > test_output_existing_tests.json
$ head -n 20 test_output_existing_tests.json # avoid showing too many lines
{{
    "test_module1.py::TestClass::test_method1": "passed",
    "test_module2.py::test_function2": "passed",
    ...
```

7. **Remove relevant code files or chunks**: Based on your exploration in step 2, introduce bugs by removing hunks or directly deleting all content from at
↪ least {min_changed_files} code files.
8. Run the original test command again and make sure some tests fail, meaning the removal breaks some functionality. Don't introduce additional syntax
↪ errors that make all tests fail. Still, redirect the output to a log file (e.g., `bash test_script.sh > test_output_bug_code.log 2>&1`) to view the
↪ results. Also, verify that your parser script can correctly parse the test results log and summarize which tests passed or failed. For example:

```bash
$ cat test_output_bug_code.log | python3 parse_test_output.py > test_output_bug_code.json
$ cat test_output_bug_code.json | grep "failed"
...
```

9. **Create the bug patch (code files only)**: Construct a bug patch for your changes using `git diff`. For example, `git diff > bug_patch.diff`. Because
↪ `git diff` won't apply to files in the index / untracked files, make sure you correctly create the bug patch (either stage all the changes and then use
↪ `git diff --cached`, or use `git diff` for tracked files). Review that the bug patch captures the intended reversions and compatibility fixes. **Verify
↪ that the patch ONLY contains changes to code files, NOT test files.**
10. **Remove/weaken tests (ONLY test files can be modified)**: Now and only now, you can modify test files. Delete entire test functions, files or remove /
↪ weaken some of the test cases that would catch your bug, creating a "test gap" where some bugs can hide. **CRITICAL: DO NOT comment out the original
↪ tests as this leaves obvious hints; instead, simply delete them or weaken assertions.**
11. **Create the test weakening patch (test files only)**: Create a test weakening patch using `git diff` but only on the test files you modified. **Verify
↪ that this patch ONLY contains changes to test files, NOT code files.**

### IMPORTANT REQUIREMENTS

1. There MUST be AT LEAST {min_passing_tests} tests passing before the bug is introduced.
2. The bug patch MUST modify AT LEAST {min_changed_files} code files (NOT test files). Modification includes adding, removing, or editing files.
3. You MUST NOT leave any hints that reveals your bug injection intention. DO NOT leave comments like "introduce bug here" in your patches. DO NOT leave the
↪ original correct code in comments.
4. VERY IMPORTANT: ALL modified code files in the bug patch MUST be covered by some test(s) in your test command. There must be NO orphan code files that are
↪ modified but not exercised by any of the selected tests.

### Required files to submit

1. test_files.txt: A text file listing all the test files you selected in step 2 to validate (1) the original code correctness and (2) the bug exposure after
↪ code removal (one unique **relative** file path per line).
2. test_script.sh: A bash script specifying how to run the tests
3. parse_test_output.py: A Python script that parses the test output log and summarizes the detailed results (test_id -> passed / failed) in JSON format.
4. bug_patch.diff: A git diff patch that introduces the bug into the code.
5. test_patch.diff: A git diff patch that removes/weakens tests to hide the bug.

### Submission format

```
<tool: submit>
test_files.txt
test_script.sh
parse_test_output.py
bug_patch.diff
test_patch.diff
</tool>
```

I've uploaded the corresponding code repository at {repo_root} and installed all the necessary dependencies. Now, the bash session has started, with the
↪ current working directory set to the repo root.

---

## History-aware bug-injection

You are working with a random commit from a code repository. Your goal is to introduce **realistic bugs** into the codebase by selectively reverting code
↪  changes from git history and applying minimal compatibility fixes to ensure the code remains runnable. The bugs will serve as training data for a
↪  bug-fixing AI system.

The bug introduction process involves two key steps:
1. **Selectively revert code changes from history**: Use git history to identify and revert specific bug fixes or improvements. You can revert entire files
↪  to historical versions, cherry-pick specific line ranges from previous commits, or combine reversions from multiple commits across multiple files. This
↪  gives you fine-grained control over bug introduction.
2. **Apply minimal compatibility fixes**: Make only the necessary adjustments to resolve trivial issues (e.g., import errors, renamed functions, API
↪  changes) so the code runs without syntax errors, while preserving the historical bugs.

### Steps to follow

1. Understand the codebase, its functionalities, and the test suite / framework / command structure.
2. **Browse git history to identify revertible changes**: Use `git log`, `git log --oneline`, `git show`, `git diff`, `git log -p`, and `git log -L
↪  <start>,<end>:<file>` (for line-range history) to explore the repository's history. Look for commits that introduced bug fixes, refactorings, or
↪  improvements to core functionality that you can revert. Focus on:
   - Bug fix commits (search for keywords like "fix", "bug", "issue", "crash", "error" in commit messages)
   - Feature enhancements or optimizations that can be reverted
   - Edge case handling that can be removed

   Identify interesting changes across code files (NOT test files) that you might want to revert. These can be:
   - Entire file restorations to historical versions
   - Specific function/method changes from particular commits
   - Line-range reversions using `git show <commit>:<file>` and manual editing
   - Combinations of multiple partial reversions across different files

   Take detailed notes on which commits, files, and line ranges contain revertible changes.
3. **Identify related tests**: Based on the code changes you've identified in step 2, find the corresponding test files that exercise those code paths. You
↪  can use `git log` on test files, search for imports/references, or run tests to see which ones are related. Select a test suite that includes at least
↪  {min_passing_tests} tests related to your target code files. The tests can also be indirectly related if you cannot find enough direct tests.
4. **Set up the test command**: Create a test command that runs your selected tests. Make sure your test command can output the detailed test results,
↪  including which tests passed or failed (e.g., `pytest -rA`); ensure that the test execution takes less than 90 seconds. Dump the test command in a bash
↪  script (e.g., `test_script.sh`, which may involve additional setups like environment variables) for later use.
5. Trigger the test command, **directing the output to a log file** (e.g., using `bash test_script.sh > test_output_existing_tests.log 2>&1`) because the
↪  test output can be very long. View the log file (e.g., using `head` or `tail`) to verify the results.
6. If not all tests pass, it may be due to flakiness or environment issues; just exclude them from the test command or find another set until all selected
↪  tests pass (still needs to satisfy the minimum passing tests requirement: >= {min_passing_tests}).
7. Write a generic parser script in Python that can parse any test log file output by your test command. The test script should read from stdin the test log
↪  output and writes to stdout a JSON summary mapping each executed test case to either "passed" or "failed". Verify that your parser works. Example usage
↪  and output:

```bash
$ cat test_output_existing_tests.log | python3 parse_test_output.py > test_output_existing_tests.json
$ head -n 20 test_output_existing_tests.json # avoid showing too many lines
{{
    "test_module1.py::TestClass::test_method1": "passed",
    "test_module2.py::TestClass::test_function2": "passed",
    ...
```

8. **Selectively revert code changes (NO TEST FILES)**: Based on your exploration in step 2, introduce bugs by reverting changes to at least
↪  {min_changed_files} code files (NOT test files). You have multiple strategies available:

   **Strategy A: Full file restoration**
   - Restore entire files to historical versions using `git show <commit>:path/to/file > path/to/file` or `git restore --source=<commit> --worktree --
   ↪  path/to/file`

   **Strategy B: Cherry-pick specific changes**
   - Use `git show <commit>` to view specific bug fixes
   - Use `git show <commit>:<file>` to see how specific files looked at that commit
   - Use `git log -L <start>,<end>:<file>` to see the history of specific line ranges
   - Manually revert just the relevant portions of those fixes by editing files

   **Strategy C: Combine multiple reversions**
   - Revert different changes from different commits across multiple files
   - Create complex multi-file bugs by reverting related changes in dependent components
   - Mix full file restorations with partial cherry-picked reversions

   **Examples of history-viewing commands:**
   ```bash
   # View a specific commit's changes
   git show <commit_hash>

   # View how a file looked at a specific commit
   git show <commit_hash>:path/to/file

   # View history of specific line range in a file (at most recent <N> commits)
   git log -L 10,20:path/to/file -n <N>

   # Restore entire file to historical version
   git show <commit_hash>:path/to/file > path/to/file

   # View diff between two commits for a file
   git diff <old_commit>..<new_commit> -- path/to/file
   ```

   Choose the strategy or combination of strategies that creates the most realistic bugs. The key is to revert actual bug fixes or improvements that were
   ↪  previously made, not to introduce arbitrary syntax errors.

```
         **CRITICAL: You MUST NOT modify or restore any test files in this step - only code files!**
    9. **Apply minimal compatibility fixes (ONLY to code files)**: After reverting code changes, apply minimal compatibility fixes to resolve only the trivial
↪    issues that prevent the code from running (e.g., import errors, renamed functions, API changes). Make only the necessary adjustments to ensure tests can
↪    execute without syntax errors, while preserving the reverted bugs. **Do NOT modify any test files in this step.** Then run the original test command
↪    again and make sure some tests fail, meaning the reverted bugs would be caught by the original tests. Don't introduce additional syntax errors that make
↪    all tests fail. Still, redirect the output to a log file (e.g., `bash test_script.sh > test_output_buggy_code.log 2>&1`) to view the results. Also,
↪    verify that your parser script can correctly parse the test results log and summarize which tests passed or failed. For example:

```bash
$ cat test_output_buggy_code.log | python3 parse_test_output.py > test_output_buggy_code.json
$ cat test_output_buggy_code.json | grep "failed"
...
```

    10. **Create the bug patch (code files only)**: Construct a bug patch for your changes (reverted changes + minimal compatibility fixes) using `git diff`.
↪    For example, `git diff > bug_patch.diff`. Because `git diff` won't apply to files in the index / untracked files, make sure you correctly create the bug
↪    patch (either stage all the changes and then use `git diff --cached`, or use `git diff` for tracked files). Review that the bug patch captures the
↪    intended reversions and compatibility fixes. **Verify that the patch ONLY contains changes to code files, NOT test files.**
    11. **Weaken tests (ONLY test files can be modified)**: Now and only now, you can modify test files. Remove or weaken some of the tests that would catch your
↪    bug, creating a "test gap" where some bugs can hide. You can remove test cases, weaken assertions, remove edge case coverage that exposes the bug, or
↪    simply reverting the test to a historical version. You don't need to hide ALL test failures; it's acceptable if some tests still fail after weakening.
↪    **CRITICAL: In this step, you MUST ONLY modify test files. Do NOT modify any code files.** Once done, run the test command again and verify that fewer
↪    tests fail compared to step 9. Still redirect the output to a log file (e.g., `bash test_script.sh > test_output_weakened_tests.log 2>&1`) to view the
↪    results and verify the parser script works as expected. **CRITICAL: DO NOT comment out failing tests as this leaves obvious hints; instead, simply
↪    delete them or weaken assertions.**
    12. **Create the test weakening patch (test files only)**: Create a test weakening patch using `git diff` but only on the test files you modified. **Verify
↪    that this patch ONLY contains changes to test files, NOT code files.**

### IMPORTANT REQUIREMENTS

1. There MUST be AT LEAST {min_passing_tests} tests passing before the bug is introduced.
2. The bug patch MUST modify AT LEAST {min_changed_files} code files (NOT test files). Modification includes adding, removing, or editing files.
3. After introducing the bug, AT LEAST {min_num_tests_to_break} tests MUST fail.
4. You MUST NOT leave any hints that reveals your bug injection intention. DO NOT leave comments like "introduce bug here" in your patches. DO NOT leave the
↪    original correct code in comments.
5. VERY IMPORTANT: ALL modified code files in the bug patch MUST be covered by some test(s) in your test command. There must be NO orphan code files that are
↪    modified but not exercised by any of the selected tests.
6. The bug MUST be realistic, something that can naturally occur in a code project. DO NOT introduce syntax errors or undefined variables that make all tests
↪    fail.

### Required files to submit

1. test_files.txt: A text file listing all the test files you selected in step 2.
2. test_script.sh: A bash script specifying how to run the tests
3. parse_test_output.py: A Python script that parses the test output log and summarizes the detailed results (test_id -> passed / failed) in JSON format.
4. bug_patch.diff: A git diff patch that introduces the bug into the code.
5. test_patch.diff: A git diff patch that removes/weakens tests to hide the bug.

### Submission format

<tool: submit>
test_files.txt
test_script.sh
parse_test_output.py
bug_patch.diff
test_patch.diff
</tool>

I've uploaded the corresponding code repository at {repo_root} and installed all the necessary dependencies. Now, the bash session has started, with the
↪    current working directory set to the repo root.
```

## Direct bug-injection

```
You are working with a random commit from a code repository. Your goal is to introduce bugs into the codebase.

### Steps to follow

1. Understand the codebase, its functionalities, and the test suite / framework / command structure.
2. **Identify interesting test files**: find a set of test files that cover significant functionality in the codebase, making sure they involve at least
↪    {min_passing_tests} tests and test over {min_changed_files} code files.
3. **Set up the test command**: Create a test command that runs your selected tests. Make sure your test command can output the detailed test results,
↪    including which tests passed or failed (e.g., `pytest -rA`); ensure that the test execution takes less than 90 seconds. Dump the test command in a bash
↪    script (e.g., `test_script.sh`, which may involve additional setups like environment variables) for later use.
4. Trigger the test command, **directing the output to a log file** (e.g., using `bash test_script.sh > test_output_existing_tests.log 2>&1`) because the
↪    test output can be very long. View the log file (e.g., using `head` or `tail`) to verify the results.
5. If not all tests pass, it may be due to flakiness or environment issues; just exclude them from the test command or find another set until all selected
↪    tests pass (still needs to satisfy the minimum passing tests requirement: >= {min_passing_tests}).
6. Write a generic parser script in Python that can parse any test log file output by your test command. The test script should read from stdin the test log
↪    output and writes to stdout a JSON summary mapping each executed test case to either "passed" or "failed". Verify that your parser works. Example usage
↪    and output:

```bash
$ cat test_output_existing_tests.log | python3 parse_test_output.py > test_output_existing_tests.json
$ head -n 20 test_output_existing_tests.json # avoid showing too many lines
{{
    "test_module1.py::TestClass::test_method1": "passed",
    "test_module2.py::test_function2": "passed",
    ...
```
```

```
7. **Introduce bugs**: Based on your exploration in step 2, introduce bugs to at least {min_changed_files} code files.
8. Run the original test command again and make sure some tests fail, meaning the bug breaks some functionality. Don't introduce trivial syntax errors that
↪   make all tests fail. Still, redirect the output to a log file (e.g., `bash test_script.sh > test_output_bug_code.log 2>&1`) to view the results. Also,
↪   verify that your parser script can correctly parse the test results log and summarize which tests passed or failed. For example:

```bash
$ cat test_output_bug_code.log | python3 parse_test_output.py > test_output_bug_code.json
$ cat test_output_bug_code.json | grep "failed"
...
```

9. **Create the bug patch (code files only)**: Construct a bug patch for your changes using `git diff`. For example, `git diff > bug_patch.diff`. Because
↪   `git diff` won't apply to files in the index / untracked files, make sure you correctly create the bug patch (either stage all the changes and then use
↪   `git diff --cached`, or use `git diff` for tracked files). Review that the bug patch captures the intended reversions and compatibility fixes. **Verify
↪   that the patch ONLY contains changes to code files, NOT test files.**
10. **Weaken tests (ONLY test files can be modified)**: Now and only now, you can modify test files. Weaken / remove some of the test cases that would catch
↪   your bug, creating a "test gap" where some bugs can hide. **CRITICAL: DO NOT comment out the original tests as this leaves obvious hints; instead, simply
↪   delete them or weaken assertions.**
11. **Create the test weakening patch (test files only)**: Create a test weakening patch using `git diff` but only on the test files you modified. **Verify
↪   that this patch ONLY contains changes to test files, NOT code files.**

### IMPORTANT REQUIREMENTS

1. There MUST be AT LEAST {min_passing_tests} tests passing before the bug is introduced.
2. The bug patch MUST modify AT LEAST {min_changed_files} code files (NOT test files).
3. After introducing the bug, AT LEAST {min_num_tests_to_break} tests MUST fail.
4. You MUST NOT leave any hints that reveals your bug injection intention. DO NOT leave comments like "introduce bug here" in your patches. DO NOT leave the
↪   original correct code in comments.
5. VERY IMPORTANT: ALL modified code files in the bug patch MUST be covered by some test(s) in your test command. There must be NO orphan code files that are
↪   modified but not exercised by any of the selected tests.

### Required files to submit

1. test_files.txt: A text file listing all the test files you selected in step 2
2. test_script.sh: A bash script specifying how to run the tests
3. parse_test_output.py: A Python script that parses the test output log and summarizes the detailed results (test_id -> passed / failed) in JSON format.
4. bug_patch.diff: A git diff patch that introduces the bug into the code.
5. test_patch.diff: A git diff patch that removes/weakens tests to hide the bug.

### Submission format

<tool: submit>
test_files.txt
test_script.sh
parse_test_output.py
bug_patch.diff
test_patch.diff
</tool>

I've uploaded the corresponding code repository at {repo_root} and installed all the necessary dependencies. Now, the bash session has started, with the
↪   current working directory set to the repo root.
```

## D.2. Bug Solving

For bug-solving with our solver agent, we instantiate the issue description with the following fixed prompt template:

---

**Fixed-template issue description specified by the oracle test patch**

```
I am improving the test suite of the project with the following changes, but the current code does not pass the tests. Please fix the code. If any existing
↪   tests relevant to the functionality being changed are failing, please make sure your patch passes those tests as well.

```diff
{oracle_test_patch}
```
```

---

We then use the following prompt for general issue solving, including both the real issues from evaluation and the fixed-template issues above:

---

**Prompt template for bug-solving**

```
Solve the following issue by implementing the necessary code changes and submitting a patch file:

<issue_description>
{issue}
</issue_description>

The [result] argument of <tool: submit> should be the path to a patch file that resolves the issue. This file must be accessible from the current working
↪   directory and should contain the end-to-end code changes in git diff format. You can refine and submit your patch multiple times as needed to ensure
↪   correctness.

Once you've submitted at least once, provide a brief summary.
```

```
Again, if you have enough budget, you should try to fully utilize it by doing more testing, checking edge cases, or even considering alternative solutions.
↪   This will help you gain more confidence in your submission. DO AS MUCH TESTING AS POSSIBLE WHENEVER YOU HAVE BUDGET. The testing involves developing new
↪   tests, confirming no existing tests are broken, and checking edge cases. Only submit when all the relevant tests pass!!

I've uploaded the corresponding code repository at {repo_root} and installed all the necessary dependencies. Now, the bash session has started, with the
↪   current working directory set to the repo root.
```

# E. Discussion

### E.1. Theoretical Analysis

With a focus on the challenger (bug-injector), we analyze the behavior of reward designs such as Equation (1) and the theoretical optimal play of the challenger-solver game in §A. We show that the challenger has several dominant strategies that can stop self-play from progressing and then discuss practical mitigations that may address these problems, some of which were adopted by this work. Our analysis help clarify what we can expect in such self-play and what must be avoided.

### E.2. Limitations

While Self-play SWE-RL demonstrates promising results toward training superintelligent software agents, it is still an early step and several limitations warrant discussion:

- **No hidden oracles**: Providing the complete oracle (i.e. the tests) in the task specification prompt may enable agents to develop reward-hacking behaviors (e.g., overfitting the specific set of tests) rather than genuine bug-fixing capabilities. Although this is not observed in this paper, future iterations should explore enriching the bug artifact with public and private test splits.

- **Unit tests for verification**: The current framework exclusively uses unit tests as the verification oracle, which represents only a subset of real-world software engineering activities. A higher-level abstraction, such as a goal, emphasized by CodeClash (Yang et al., 2026b), may serve as a more scalable way to verify software correctness.

- **Lack of model variants and role separation:** Our experiments use a single model configuration for both roles. Future work should explore larger Mixture-of-Experts (MoE) models to understand how capacity and architecture affect self-play learning, and consider separate policies for each role.

### E.3. Unsuccessful Attempts

We share our setbacks here for insight, but this does not imply they cannot work in the future:

- **Synthesizing natural language issues in self-play**: Our approach focuses on formal test specifications instead of natural language issues. While this design minimizes data assumptions and proves effective for learning, our initial attempts failed to reliably generate high-quality and unambiguous issue descriptions. The generated issues tend to copy test patches, are logically incoherent, and collapse to identical patterns. We attribute this to limited natural language capabilities of our 32B base model (CWM) and the opaque reward signals that fail to promote quality or diversity.

- **Repository-specialized training**. Real-world development often involves sustained work within specific codebases where deep contextual knowledge provides significant advantages. We explored training repository-specialized agents by performing SSR on just 23 repo images from SWE-bench Verified and SWE-Bench Pro, preventing leakage by selecting commits prior to any evaluation instances. However, this did not outperform training on more non-overlapping repos, likely because 23 repos provide insufficient diversity in problem-solving patterns.

- **Stable training at scale**: Despite following (Devvrit et al., 2026; Zheng et al., 2025), we still observed training instability that prevents SSR from further scaling, manifesting as gibberish outputs. This instability may stem from a combination of suboptimal hyperparameter settings, intrinsic challenges of long-horizon rollouts, model pretraining dynamics, and fundamental properties of self-play learning. Addressing this instability is critical to understanding how self-play scales and to unlocking its full potential.

### E.4. Future Work

Besides addressing the aforementioned limitations, our work opens several promising research directions for advancing self-play learning and superintelligence in software engineering agents:

- **Distribution control with seeding:** The current approach lacks explicit control over bug injection locations, which can lead to duplicate bugs or distribution bias when sampling multiple times from the same repository. To improve diversity, future work could adopt seeding techniques from Magicoder (Wei et al., 2024), providing the bug-injection agent with contextual information such as target code snippets or specific files to guide the generation process toward diversity.

- **Synthesizing complex multi-step software tasks:** While our approach handles repository-level bug fixing, many real-world tasks require complex, multi-step workflows, such as major version migrations or building new software stacks from scratch. Our concept of higher-order bugs is a first step, introducing layered dependencies between bug injection and repair. Enabling such tasks will require new scaffolding designs, such as multi-context rollouts spanning interdependent sessions and automated context compaction (Anthropic, 2025; OpenAI, 2025).

- **Efficient training paradigms for long-horizon software agents:** Real-world software projects, such as building a production-grade RL codebase, pose unique challenges for agent training: they span months of iterative development, involve thousands of interdependent decisions, and cannot be fully validated by unit tests alone. Current outcome-based RL is inefficient for such tasks because sparse terminal rewards provide little signal when most trajectories fail, making credit assignment across thousands of steps intractable. Addressing this requires new paradigms that exploit the asymmetry between generation and verification (Wei, 2025) for self-verification while delivering dense and structured feedback beyond scalar rewards.

