# OpenReview forum: "Toward Training Superintelligent Software Agents through Self-Play SWE-RL"
_ICML.cc/2026/Conference — ICML 2026 regular_

### Official Review · Reviewer_ssFQ · 2026-03-02

**Soundness:** 3
**Presentation:** 4
**Significance:** 3
**Originality:** 3
**Overall Recommendation:** 4
**Confidence:** 5

**Summary:**

This paper introduces Self-play SWE-RL (SSR), a framework to train large language models (LLMs) as software engineering agents under minimal data assumptions (requiring only source code and dependency sandboxes). A single LLM plays two interacting roles: a bug-injection agent and a solver agent. Trained via reinforcement learning in a self-play setting, SSR demonstrates consistent self-improvement on SWE-bench Verified and SWE-Bench Pro, outperforming baselines trained on human-curated data.

**Compliance With Llm Reviewing Policy:**

Affirmed.

**Key Questions For Authors:**

1. Given that recent works scale up static/real data to achieve 64.0% with 30B models, and the paper explicitly criticizes static generation methods like SWE-smith, why are there no direct empirical comparisons with these methods? How can the authors definitively attribute SSR's gains to the "self-play" dynamics itself rather than merely the increased volume of generated training data?

2. Given that SSR completely abandons natural language issues during training, yet evaluations (SWE-bench) heavily rely on them, how do the authors explain the zero-shot generalization? Does this suggest the model only improved its pure coding ability rather than its requirement comprehension?

3. Since the experiments are limited to the CWM model, how do the authors justify the cross-model generalizability of the SSR framework?

4. Since the solver agent has full access to the oracle tests in the prompt, how does the current framework practically prevent test-overfitting or reward hacking?

**Limitations:**

Yes.
The authors have thoroughly discussed limitations in the appendix, including the lack of hidden oracles , reliance on unit tests , and the failure to synthesize natural language issues. It is recommended that the authors explicitly acknowledge the lack of empirical comparisons with static synthetic pipelines and the generalizability risks of evaluating on a single model.

**Strengths And Weaknesses:**

Soundness & Significance:
- Strengths: The experimental design is rigorous, comparing SSR directly against a human-data baseline. The ablation studies effectively validate the necessity of the joint (injection + repair) self-play mechanism.
- Weaknesses:
  - Lack of Empirical Comparison with Concurrent Synthetic Pipelines: The authors criticize static synthetic bug generation pipelines (e.g., SWE-smith, BugPilot) for their stronger assumptions and inability to provide continuous self-improvement. However, the paper provides zero empirical comparisons against these static methods. Furthermore, recent work like "Immersion in the GitHub Universe: Scaling Coding Agents to Mastery" achieves a 64.0% resolve rate using a 30B model by simply scaling up static data, vastly outperforming this paper's 51.4%. This raises a fundamental question: is the highly complex and unstable self-play mechanism merely over-engineering, with its purported advantages remaining purely theoretical compared to simple data scaling?
  - Misalignment between Training and Evaluation: The framework relies entirely on formal reversed test patches as task specifications during training and abandons natural language issues (an attempt the authors admit failed). This creates a stark disconnect with the SWE-bench evaluation paradigm and real-world scenarios, significantly weakening the claims of practical utility.
  - Questionable Generalizability: Experiments are exclusively conducted on a single base model (CWM-sft). It remains highly uncertain whether other models would yield similar improvements.
  - Relying exclusively on unit tests as the verification oracle theoretically opens the door to test-overfitting.

Presentation & Originality:
- Strengths: The mechanism of automatically constructing complex bug artifacts via "code removal + history reversion + test-weakening" is highly creative. Transparently documenting "unsuccessful attempts" and training instabilities in the appendix is commendable.
- Weaknesses: Using synthetic data for agent training is not entirely new (e.g., SWE-smith). While framing this as a "dynamic self-play game" offers a novel perspective, this conceptual originality severely lacks empirical backing to prove its superiority over existing static pipelines.

---

> ### Author Rebuttal · Authors · 2026-03-31
>
> Dear Reviewer ssFQ, we deeply appreciate your detailed and constructive review. In our responses below, we address each primary question (denoted as Q) and comment (denoted as C). Should there be any misunderstandings of your concerns, please kindly let us know; we are eager to communicate throughout the discussion period.
>
> > Q1: Given that recent works scale up static/real data to achieve 64.0% with 30B models…
>
> Great point. Please kindly note that we already compare against a real-data RL baseline that uses labeled issue descriptions and test commands. For synthetic data methods like SWE-smith, they typically rely on teacher models for distillation and stronger data assumptions such as access to human-specified test suites and parsers [1], while our focus is **self-improvement from raw repositories without teacher-generated trajectories**. The repository coverage is also different from ours, so directly comparing with their data would not be fair under our setting.
>
> In the paper, we already include a fairer comparison through the repair only setting in Figure 9a. In this setting, we replace the dynamic proposer update with training on a fixed pool of valid self-generated bugs. The table below summarizes these key results on the combined SWE-bench Verified + Pro benchmark, totaling 1231 instances.
> |Variant|Resolve rate|
> |-|-|
> |Baseline RL|34.9%|
> |Repair only setting|33.5%|
> |Self-play|38.0%|
>
> It shows that full self-play is clearly better than the other settings. Increased data volume is a natural consequence of self-play and contributes to the effectiveness, but the dynamic proposer-solver loop makes the pipeline more effective than static training on a fixed bug pool.
>
> [1] Yang et al. SWE-smith: Scaling Data for Software Engineering Agents. NeurIPS'25
>
> > Q2: Given that SSR completely abandons natural language issues during training…
>
> Great question. We view software engineering tasks as involving two ingredients: understanding the natural-language requirement, and reasoning over the repository to produce a correct fix. The second part already includes difficult phases such as navigating the repository, localizing the bug, planning edits, and validating candidate patches [2]. Our setup directly trains this repository-level reasoning, while requirement interpretation becomes understanding executable test specifications instead of natural language. The remaining transfer is therefore at the natural-language interpretation layer, which is already supported by standard pretraining on large amounts of natural-language data.
>
> So we do not interpret the result as showing that natural language is unimportant. Rather, the paper shows that strong repository-level reasoning and bug-fixing ability can be improved through executable feedback alone and then transfer to natural-language issue resolution. As discussed in Sec. F.4, handling natural language more directly is an important next step.
>
> [2] Xia et al. Agentless: Demystifying LLM-based Software Engineering Agents. FSE'25
>
> > Q3: Since the experiments are limited to the CWM model…
>
> We acknowledge this limitation. As discussed in Sec. F.2, extending SSR to larger model scales and separate policies for each role is important future work.
>
> > Q4: Since the solver agent has full access to the oracle tests in the prompt…
>
> We address reward hacking through several concrete mechanisms. First, `.git` is removed and the repository is reinitialized, which blocks leakage through repository history [3]. Second, bug artifacts must pass strict consistency validation before entering training. Third, the solver is evaluated only after restoring the original oracle tests, so editing tests cannot directly improve the reward.
>
> To prevent test-overfitting, we primarily rely on the model prior to produce a general test-passing fix after understanding the requirements, rather than overfitting to the oracle tests. However, this is not fully prevented in the current setup, but we did not observe it in the paper. As discussed in Sec. F.2 under "No hidden oracles," stronger defenses such as hidden or public/private oracle splits are an important next step.
>
> [3] Repo State Loopholes During Agentic Evaluation. https://github.com/SWE-bench/SWE-bench/issues/465
>
> > C1: Using synthetic data for agent training is not entirely new…
>
> We want to kindly mention that the main novelty is **self-improvement from raw repositories minimal data assumption (e.g., no human-authored issue descriptions or test commands)**, rather than synthetic data alone. As discussed in Q1, methods such as SWE-smith rely on teacher models and stronger data assumptions, while our setting does not. We also compare against the repair only setting in Figure 9a, and full self-play is better.
>
> ---
>
> We hope these clarifications make the paper's scope and evidence more precise. If these points address the main concerns, we would appreciate reconsideration of the overall score.

---

> > ### Author Rebuttal · Reviewer_ssFQ · 2026-04-02
> >
> > Thank you for the detailed rebuttal. While I appreciate the clarifications regarding your baseline comparisons and the internal ablations, my core concerns remain. Specifically, the performance improvements (a ~3.1% gain) seem relatively modest given the extreme complexity of the self-play RL system. Furthermore, the fact that experiments were conducted on only a single model raises questions about generalizability, and the complete disconnect from natural language during training remains a critical limitation for SWE tasks. Given these factors, I am inclined to maintain my current score.

---

> > > ### Author Response · Authors · 2026-04-08
> > >
> > > Thank you again for the detailed and thoughtful review. We appreciate your **positive assessment of several parts of the paper, including the rigor of the human-data baseline comparison, the ablations supporting the joint self-play mechanism, and the creativity of the executable bug-artifact construction**. We are also grateful for your overall supportive assessment despite the limitations you identified.
> > >
> > > Your remaining concerns are well taken, and in the revision we will make the scope much more precise. In particular, we will explicitly avoid implying that this paper shows self-play is broadly superior to scaled static pipelines. We will also foreground the natural-language mismatch, single-model scope, and oracle-test limitations more clearly.
> > >
> > > Finally, we want to reiterate the intended claim: **not SOTA over all alternatives, but that dynamic self-play over raw repositories can produce useful learning signals and drive self-improvement under minimal data assumptions, without human-authored issues or tests.**

---

### Official Review · Reviewer_XaCJ · 2026-03-10

**Soundness:** 2
**Presentation:** 3
**Significance:** 3
**Originality:** 3
**Overall Recommendation:** 4
**Confidence:** 4

**Summary:**

This paper presents Self-play SWE-RL (SSR), a reinforcement learning framework for training software engineering agents under minimal data assumptions. The manuscript's central domain comprises self-play RL applied to code repositories, where a single LLM policy alternates between two roles: a bug-injection agent that introduces bugs and weakens tests to hide them, and a bug-solving agent that repairs bugs using only the reversed test-weakening patch as a formal task specification — requiring no human-curated issue descriptions or test suites. Overall, the authors analyze a central concept of grounding self-play in real-play codebases to generate an evolving bug curriculum, demonstrating consistent improvements over both the base model and a human-data RL baseline on SWE-bench Verified and SWE-Bench Pro throughout training.

**Compliance With Llm Reviewing Policy:**

Affirmed.

**Key Questions For Authors:**

1. The SSR vs. Baseline RL gap on SWE-bench Verified (2.4%) falls within the reported evaluation noise (~2%). Were multiple independent runs conducted? Can the authors provide confidence intervals or significance tests to rule out random variation as a primary explanation?

2. The number of repositories and total bugs generated are not reported. How does performance scale with the number of repositories? Is there evidence of diminishing returns, and what does this imply about the practical upper bound of the approach?

3. The authors theoretically identify dominant challenger strategies that could halt learning progress. Were any signs of injector behavior collapse or diversity reduction observed during training? Is the consistency validation mechanism sufficient to prevent such degeneration in practice?

**Limitations:**

see weaknesses and questions

**Strengths And Weaknesses:**

**Strengths**

- Training software engineering agents via self-play grounded in real-world codebases is a novel and promising direction. By having a single model alternate between proposing and solving bugs entirely from raw repositories, SSR demonstrates that meaningful learning signals can be generated without relying on explicit human supervision at training time. The fact that a model trained purely on self-generated formal specifications can generalize to human-written natural language issues at evaluation time further validates the paradigm's broader potential.

- Unlike static bug generation pipelines (e.g., SWE-smith, BugPilot), SSR's training distribution evolves online alongside the model's improving capability — as the solver gets stronger, the injector is incentivized to generate harder bugs. This automatic curriculum effect is a practically important property, as it means the training signal remains informative throughout the entire training trajectory rather than becoming stale as the model improves.

- The authors proactively analyze the challenger's dominant strategies in Appendix A, acknowledging the theoretical ceiling of self-play and discussing practical mitigations.

**Weaknesses**

- The performance gap between SSR and Baseline RL on SWE-bench Verified is only 2.4 points, while the authors themselves report evaluation noise of ~2%. Without multiple independent runs, confidence intervals, or significance tests, it is difficult to rule out random variation as a contributing explanation — the visual consistency of training curves, while suggestive, is not a substitute for proper statistical validation.

- The number of repositories used and bugs generated is not reported in detail, making it challenging to assess scaling behavior, reproducibility, or where the method's practical ceiling lies. A more thorough characterization of the training data distribution would strengthen the empirical claims.

- All experiments use one 32B model with shared parameters for both roles. Whether self-play remains effective across different model scales or with role-separated policies is left unexplored, limiting the generalizability of the findings.

---

> ### Author Rebuttal · Authors · 2026-03-31
>
> Dear Reviewer XaCJ, we deeply appreciate your thoughtful review and constructive suggestions. In our responses below, we address each primary question (denoted as Q) and comment (denoted as C). Should there be any misunderstandings of your concerns, please kindly let us know; we are eager to communicate throughout the discussion period.
>
> > Q1: The SSR vs. Baseline RL gap on SWE-bench Verified (2.4%) falls within the reported evaluation noise (~2%)…
>
> Great question. We view the final 2.4-point Verified gap as one part of a broader result. We evaluate one checkpoint every 10 steps, using a different randomized evaluation seed each time, and we do not report repeated evaluations of a fixed checkpoint because a full Verified+Pro evaluation run already costs about **413 H100 GPU-hours**.
>
> More importantly, the paper's main contribution is **self-improvement from raw repositories without human-authored issue descriptions or test commands**. As shown in the paper, SSR improves steadily and reaches performance comparable to the RL baseline requiring human-authored or human-curated issue descriptions plus the corresponding test suites and test commands.
>
> To further test whether the gains could be explained by evaluation noise, we performed additional paired analysis on the same benchmark instances. The final combined resolve rates over 1231 instances are 38.0% for SSR and 34.9% for baseline RL. The table below counts how often the later or stronger system solves an instance that the earlier or weaker system misses, versus the reverse. McNemar's test is a standard paired test for this imbalance [1].
> |Comparison|Fail to pass|Pass to fail|McNemar p|
> |-|-|-|-|
> |SSR vs baseline RL|121|83|$9.6\times10^{-3}$|
> |SSR step 10 to 150|147|63|$6.5\times10^{-9}$|
>
> In both rows, fail-to-pass clearly exceeds pass-to-fail, and the low p-values indicate that the observed gains are statistically significant under this paired test.
>
> [1] McNemar. Note on the sampling error of the difference between correlated proportions or percentages
>
> > Q2: The number of repositories and total bugs generated are not reported…
>
> Great question. The training data statistics are summarized in the table below. We train on 2624 unique repositories, encompassing 12371 unique labeled issues (only used for baseline RL). During SSR training, the injector generates 33370 bug instances in total.
> |Statistic|Value|
> |-|-|
> |Unique repositories|2624|
> |Unique labeled issues|12371|
> |Total generated bugs|33370|
>
> Regarding repository scaling and diminishing returns, as discussed in Sec. F.3, we explored training repository-specialized agents by performing SSR on just 23 repo images from SWE-bench Verified and SWE-Bench Pro. This yields much less improvement than the normal setup, which suggests that repository count and diversity matter. However, we do not yet have a controlled repo-count scaling law, which is an important direction for future work. We will add the above statistics and the scaling discussion to the revision.
>
> > Q3: …Were any signs of injector behavior collapse or diversity reduction observed during training? Is the consistency validation mechanism sufficient to prevent such degeneration in practice?
>
> It depends on the setting. As discussed in the direct-injection ablation (Sec. 3.3), the naive direct-injection variant does show this failure mode, where injected bugs collapse into superficial one-line modifications. In the full SSR setting, we do not observe it. Please also kindly refer to Reviewer iHpq Q2 for the additional analysis of difficulty and consistency over training.
> We also summarize the consistency validation mechanism below, which lets the injector construct grounded bug artifacts and is sufficient to prevent the majority of loopholes and degeneration in practice, according to our experiments.
> |Check|Role|
> |-|-|
> |Original repository must pass|Ensures a valid starting state|
> |Test files/parser/test script must run|Ensures executable supervision|
> |Injected repository must fail|Confirms a real behavioral regression|
> |Test weakening must hide some failures|Defines an executable target spec|
> |Inverse mutation testing|Verifies each modified file is causally necessary|
> |Solvability + deduplication filters|Keeps usable, non-duplicate bugs|
>
> > C1: All experiments use one 32B model with shared parameters for both roles. Whether self-play remains effective across different model scales or with role-separated policies is left unexplored, limiting the generalizability of the findings.
>
> This is a great point and we acknowledge the concern. While our main focus here is self-improvement with one shared policy, as discussed in Sec. F.2, extending SSR to larger model scales and separate policies for each role is an important future work.
>
> ---
>
> We hope the added statistical and scale evidence make the empirical scope and contribution more precise. If these points address the main concerns, we would appreciate reconsideration of the score.

---

> > ### Author Rebuttal · Reviewer_XaCJ · 2026-04-01
> >
> > Thank you for your response. Some of my concerns have been addressed (e.g., the scale of the training data). However, given certain remaining limitations of the paper — such as the relatively modest performance improvements and the fact that experiments were conducted on only a single model — I am inclined to maintain my current score (4: Weak Accept).

---

> > > ### Author Response · Authors · 2026-04-08
> > >
> > > Thank you again for your careful review and constructive discussion. We appreciate your **positive assessment of the core idea, including the novelty of grounding self-play in real-world codebases, the potential of an evolving bug curriculum, and the broader promise of learning without explicit human supervision at training time**. We are also grateful for your supportive overall assessment.
> > >
> > > We also take your remaining concerns seriously. In the revision, we will better align the framing with the empirical scope, surface the added statistical and training-data details more prominently, and more clearly acknowledge the current limitations around modest margins, single-model evaluation, and unexplored role-separated variants.
> > >
> > > Finally, we want to make the central claim more explicit: **the contribution is not broad SOTA, but evidence that self-play over raw repositories can enable self-improvement for SWE agents under substantially weaker data assumptions (e.g., without human-authored issues or test commands).**

---

### Official Review · Reviewer_iHpq · 2026-03-12

**Soundness:** 2
**Presentation:** 3
**Significance:** 3
**Originality:** 3
**Overall Recommendation:** 3
**Confidence:** 4

**Summary:**

This work trains LLM with self-play for software engineering tasks. The LLM is co-trained using RL to both propose bugs and solve them. The bug generation is tasked with removing code or reverting changes from the git history and is trained to generate bugs that are consistent and are neither too hard or easy to solve. The solver is trained to solve the generated bugs successfully. Various measures are taken to filter generated bugs and ensure quality of data. Experiments show that such a self-play setup outperforms directly RL training a solver model. Further ablations show the importance of different components of the approach.

**Compliance With Llm Reviewing Policy:**

Affirmed.

**Final Justification:**

The rebuttal addressed some queries I had. I believe in the current form of the paper, the main claims are not well justified by the experiments. The authors clarify their claims as being more tightly scoped around training from self-generated bugs. I agree that this is a promising direction of enquiry. However, the paper in its current form lacks experiments tht would thoroughly investigate this. Thus I will maintain my score.

**Key Questions For Authors:**

1. How many seeds are the evaluations being reported over. Since SWE-Bench evaluation is known to be high-variance, adding confidence intervals would help.
2. Do the bugs get harder over training iterations or simply more consistent?
3. What are the details of data used for RL baseline? Is the problem description text included?

**Limitations:**

yes

**Strengths And Weaknesses:**

The central approach of using self-play is very interesting. However evidence suggests that bug-injection is only being trained to make more consistent bugs. If bugs are not become harder / adapting over time, it undermines the central claim (of leading to "superhuman" performance). More thorough evaluation and clearer ablation of bug-injection training to clarify whether bugs are being trained to be consistent or are also adapting over time would help support the central claims.

Strengths:
- The approach of using self-play for SWE agents is novel and well motivated in the paper, especially as (1) it enables synthetic problem generation and (2) may lead to superhuman performance.
- Generating bugs synthetically can be tricky but the paper includes a variety of consistency checks to ensure data quality.

Weaknesses:
- Significance of SWE bench evaluation numbers: Evaluation of agents is often very noisy. Reporting statistics such as mean/median over multiple evaluation runs would add confidence the paper's claims, especially since the difference between self-play and RL is so small.
- Section 3.5 shows that feedback from the solver while training for bug-generation provides "only a slight and largely negligible advantage" over simply using consistency signals for bug-generation. If bug-generation can mostly be trained with consistency, the claim of self-play where the harder bugs are produced over time is weakened. It means that LLMs can be trained by bugs that they self-generate but if the bugs are not getting harder / adapting, it undermines the path to superhuman performance (as claimed by the title).

---

> ### Author Rebuttal · Authors · 2026-03-30
>
> Dear Reviewer iHpq, we deeply appreciate your careful review and thoughtful questions. In our responses below, we address each primary question (denoted as Q) and comment (denoted as C). Should there be any misunderstandings of your concerns, please kindly let us know.
>
> > Q1: How many seeds are the evaluations being reported over…
>
> We evaluate one checkpoint every 10 steps with a different randomized evaluation seed, but we do not conduct multiple evaluations of a fixed checkpoint because a full Verified+Pro sweep already costs about **413 H100 GPU-hours**. We will state that more explicitly in the revision. At the same time, the paper's main contribution does not hinge on whether the final margin over baseline RL is large. The main contribution is **self-improvement from raw repositories without human-authored issue descriptions or test commands**. As shown in the paper, SSR improves steadily and reaches performance comparable to the RL baseline requiring human-authored or human-curated issue descriptions plus the corresponding test suites and test commands.
>
> To further test whether the gains could be explained by evaluation noise, we performed additional paired analysis on the same benchmark instances. The final combined resolve rates over 1231 instances are 38.0% for SSR and 34.9% for baseline RL. The table below counts how often the later or stronger system solves an instance that the earlier or weaker system misses, versus the reverse. McNemar's test is a standard paired test for this imbalance [1].
> |Comparison|Fail to pass|Pass to fail|McNemar p|
> |-|-|-|-|
> |SSR vs baseline RL|121|83|$9.6\\times10^{-3}$|
> |SSR step 10 to 150|147|63|$6.5\\times10^{-9}$|
>
> In both rows, fail-to-pass clearly exceeds pass-to-fail, and the low p-values indicate that the observed gains are statistically significant under this paired test.
>
> [1] McNemar. Note on the sampling error of the difference between correlated proportions or percentages
>
> > Q2: Do the bugs get harder over training iterations or simply more consistent?
>
> Great question. The bugs become both more challenging and more consistent. That is exactly the behavior encouraged by the reward design. As described in the paper, easier bugs increase solver reward, harder bugs increase injector reward, and bugs that are too hard are penalized. This pushes the injector toward bugs near the frontier of the solver's current capability rather than toward either trivial or unsolvable proposals.
>
> We also performed additional analysis to evaluate these two effects directly. Following recent agent work that treats trajectory length as a key complexity axis [2], we use trajectory length as a practical proxy for difficulty. The 50-step bucket averages are below.
> |Metric|1 to 50|51 to 100|101 to 150|
> |-|-|-|-|
> |Generator traj length|47.7|62.9|65.1|
> |Solver traj length|21.5|29.9|33.4|
> |Injector pass rate|2.3%|9.3%|14.4%|
>
> The first two rows indicate that both generating and solving the bugs require longer trajectories over training, so the bugs become more challenging. The last row indicates that more generated bugs pass the injector-side validity checks over time, so they also become more consistent.
>
> [2] He et al. TRAJECT-Bench: A Trajectory-Aware Benchmark for Evaluating Agentic Tool Use. ICLR'26
>
> > Q3: What are the details of data used for RL baseline? Is the problem description text included?
>
> As described in Sec. 3.2, the RL baseline uses the same hyperparameters and identical environment images as SSR, but it additionally receives human-authored or human-curated issue descriptions together with the corresponding test suites and test commands for reward computation. So yes, the problem description text is included. Please also refer to Reviewer XaCJ Q2 for the training corpus statistics.
>
> > C1: Significance of SWE bench evaluation numbers...
>
> Please refer to Q1. We want to kindly highlight that the combined result is measured over **1231 benchmark instances**, which is already a fairly stable evaluation, and the paired McNemar test in Q1 shows that the observed gain is statistically significant. More importantly, the paper's main contribution is that **SSR shows self-improvement from raw repositories without human-authored issue descriptions or test commands**, while the RL baseline relies on human data.
>
> > C2: Section 3.5 shows that feedback from the solver while training for bug-generation provides "only a slight and largely negligible advantage"…
>
> Good point. As stated in the paper, Sec. 3.5 removes one solver-feedback term, but the training still co-evolves in a single policy and the bug distribution is still adaptive. This is different from repair-only, which trains on a static bug pool. Figure 9a shows that repair-only is clearly below full self-play. Please also refer to Q2 for the additional analysis.
>
> ---
>
> We hope these clarifications address the main concerns and would appreciate any reconsideration of the score.

---

> > ### Author Rebuttal · Reviewer_iHpq · 2026-04-04
> >
> > Thank you for your response, they clarifies some of my questions, especially around significance of score and changes in behaviour of bug generator. I still have significant concerns about the positioning of the paper as a path to superinteligent software agents. I agree that while the bug-generator's problems are getting harder, it is unclear if this will lead to the bug-generator producing novel types or diverse kinds of bugs. One possible failure mode could be saturation to a paritcular kind of bug-fixing that does not generalise. This does not detract from the observation that using the model to generate bugs for itself is a useful way to create training data in a way that matches RL on more expensive, human crafted data. This could have downstream applications to training on private repositories or settings which are data sparse. These would be valuable directions to explore. However I am not fully convinced whether continuing to train using the propsoed self-play regime would lead to superhuman swe-agents.
> >
> > Since this framing seems to be such a significant part of the paper, I am inclined towards maintaining my score.

---

> > > ### Author Response · Authors · 2026-04-08
> > >
> > > Thank you again for the thoughtful follow-up. We appreciate that you described **the central self-play approach as very interesting**, and that in your follow-up you said **our response clarified some of your questions, especially around significance of score and changes in bug-generator behavior**. We also appreciate your observation that **self-generated bugs may be a useful way to create training data**, with possible downstream **value in private-repository or data-sparse settings**.
> > >
> > > We agree with your key remaining concern: showing that bugs become harder over training is not the same as showing that long-run self-play will necessarily produce broad, diverse, or superhuman SWE capability. In the revision, we will narrow the positioning accordingly. We will make it much more explicit that diversity saturation, narrow curricula, and limited bug-family coverage are open questions rather than resolved claims.
> > >
> > > Finally, we would like to reiterate that our intended claim is more modest: **not SOTA or superhuman SWE, but that raw-repository self-play can generate useful training signal and support sustained self-improvement without human-authored issue descriptions or test commands.**

---

### Official Review · Reviewer_L1Bo · 2026-03-15

**Soundness:** 3
**Presentation:** 3
**Significance:** 2
**Originality:** 3
**Overall Recommendation:** 3
**Confidence:** 4

**Summary:**

The paper propose a self-play RL mechanism for SWE tasks (SSR) that learn through raw codebases without test cases. The method involves a bug-injection policy - that inject bugs into these codebases, and a bug-solver policy - that solve these bugs.
The bugs, after consistency check, are sent to the solver.
Both policies are the same model playing 2 roles.
Results show improvement on SWE-Bench for code world models

**Compliance With Llm Reviewing Policy:**

Affirmed.

**Final Justification:**

The rebuttal is helpful. But I keep my scores unchanged.

**Key Questions For Authors:**

NA

**Limitations:**

yes

**Strengths And Weaknesses:**

* Strengths
- The paper propose a possible solution to self-play RL without human feedback for SWE tasks.
- Results are promising for models with this size.

* Weaknesses
- Title with "super-intelligent" is misleading and arrogant, we are not at any closer to superintelligence yet.
- The method didn't have a mechanism of defense against reward hacking?
- There is no mention or mechanisms to ensure the bugs produced by bug-injector are genuine and valid.
- There are no training-time charts showing the reward-curve of the **bug-injector** and bug-solver. It might be the case that bug-injector is saturated and fail to improve, or bug-solver is not improving.
- Results are minimal and missing other baselines.

---

> ### Author Rebuttal · Authors · 2026-03-30
>
> Dear Reviewer L1Bo, we deeply appreciate your constructive feedback on our work. In our responses below, we address each primary comment (denoted as C). Should there be any misunderstandings of your concerns, please kindly let us know; we are eager to communicate throughout the discussion period.
>
> > C1: Title with "super-intelligent" is misleading and arrogant; we are not any closer to superintelligence yet.
>
> We agree that the current SSR system is not close to superintelligence, and we regret if the title gave that impression. At the same time, we respectfully think there may be a misunderstanding of the paper's intended claim. The paper does not claim that the current system is already superintelligent, but rather that self-play on raw repositories may be a viable path toward increasingly capable software agents under much weaker supervision. The empirical evidence we provide is the observed self-improvement on software tasks. We will clarify this distinction more explicitly in the revision.
>
> > C2: The method didn't have a mechanism of defense against reward hacking?
>
> We want to kindly highlight that the paper does include concrete defenses against reward hacking. First, `.git` is removed and the repository is reinitialized to prevent information leakage through history [1]. Second, the solver is evaluated only after restoring the original oracle test files, so modifying tests cannot directly improve the reward. Third, bug artifacts themselves must pass strict consistency validation before entering training. More broadly, this point is already discussed in Sec. F.2 of the paper: we explicitly note there that exposing the complete oracle may in principle enable reward hacking, while also stating that this was not observed in this paper. Importantly, we also discussed the stronger defense of public/private or hidden test splits.
>
> [1] Repo State Loopholes During Agentic Evaluation. https://github.com/SWE-bench/SWE-bench/issues/465
>
> > C3: There is no mention or mechanisms to ensure the bugs produced by bug-injector are genuine and valid.
>
> We want to kindly mention that this mechanism is already described in the paper, and restate it more explicitly here. The injector does not emit arbitrary text tasks. It must construct grounded bug artifacts through code removal or historical reversion plus test weakening, and each artifact is executed through consistency validation before training use, which is summarized in the following table:
> |Check|Role|
> |-|-|
> |Original repository must pass|Ensures a valid starting state|
> |Test files/parser/test script must run|Ensures executable supervision|
> |Injected repository must fail|Confirms a real behavioral regression|
> |Test weakening must hide some failures|Defines an executable target spec|
> |Inverse mutation testing|Verifies each modified file is causally necessary|
>
> > C4: There are no training-time charts showing the reward curve of the bug-injector and bug-solver. It might be the case that bug-injector is saturated and fails to improve, or bug-solver is not improving.
>
> To address this concern directly, we performed an additional analysis, which does not suggest saturation over the 150-step training horizon shown in the paper. We summarize the training pass rates in the table below using 50-step bucket averages:
>
> |Steps|Injector pass|Solver pass|
> |-|-|-|
> |1 to 50|2.3%|56.1%|
> |51 to 100|9.3%|65.2%|
> |101 to 150|14.4%|68.3%|
>
> The table above indicates consistent reward improvement for both roles rather than early saturation. As noted in Sec. F.3 of the paper, SSR still faces stable-training-at-scale constraints, which limit broader training-dynamics studies in the current version. We will add this discussion to the revision.
>
> > C5: Results are minimal and missing other baselines.
>
> We understand the concern, but for the paper's intended claim, the results are more meaningful than "minimal" may suggest. The goal is not broad SOTA, but showing that **self-play from raw repositories can improve steadily and reach competitive performance relative to a stronger-supervision RL baseline**. Within that scope, the gains are substantial. From the base model to step 150, the main run improves by **10.4 points** on SWE-bench Verified and **7.8 points** on SWE-Bench Pro. The paper also includes two important controls: (1) a strong human-data RL baseline with the same hyperparameters and environment images, and (2) repair-only, the closest within-paper static-data control, which underperforms full SSR. More external baselines would further strengthen the paper, but the current comparisons already test the central claim on its intended scope.
> The key combined results on SWE-bench Verified and Pro are listed below.
> |Setting|Resolve rate (%)|
> |-|-|
> |Base model|29.2|
> |Repair only|33.5|
> |Baseline RL|34.9|
> |SSR|38.0|
>
> ---
>
> We hope the clarified scope and added evidence address the main concerns and would appreciate any reconsideration of the score.

---

> > ### Author Rebuttal · Reviewer_L1Bo · 2026-04-02
> >
> > Thanks for the rebuttal. However, I think some issues remain unresolved, such as C1, and C3 and C5

---

> > > ### Author Response · Authors · 2026-04-08
> > >
> > > Thank you again for the direct and constructive feedback. We appreciate that you **recognized the paper as proposing a possible solution for self-play RL on SWE tasks without human feedback**, and that you **noted the results are promising for models of this size**. Your comments were also very helpful in showing us where the presentation needs to be more precise.
> > >
> > > We agree that the current "superintelligent" framing is too strong for the evidence in this version, and we will substantially soften this language in the revision. We will also make the bug-validity mechanisms much more explicit in the main paper: valid bugs are grounded in repository changes, the original repo must pass, the injected repo must fail, weakening must hide some failures, and inverse mutation testing checks that each modified file is causally necessary. We will also surface the reward-hacking defenses more clearly.
> > >
> > > Finally, we will clarify that **the paper’s claim is not broad SOTA**. The core claim is that **self-play on raw repositories can drive meaningful self-improvement under much weaker supervision than standard human-data RL**.

---

### Decision · Program_Chairs · 2026-04-30

**Decision:**

Accept (regular)

**Comment:**

This paper introduces Self-play SWE-RL, a novel approach that reduces reliance on human-curated data by having a single model alternately inject and repair bugs in real repositories using executable specifications.

The proposed online bug-generation and repair loop is technically compelling, and the comparison against human-data RL baselines provides a credible empirical foundation for the framework’s utility over static synthetic pipelines. However, my recommendation is capped at Weak Accept due to two main limitations. First, the paper significantly overreaches in its framing; the current evidence supports a narrow, model-specific improvement rather than the broad, “superintelligent” SWE capabilities suggested by the authors. Second, the empirical gains are modest, restricted to a single base model, and lack rigorous statistical validation. While these weaknesses do not negate the core conceptual contribution, they highlight that this is a promising first step rather than a definitive solution. I strongly advise the authors to temper their claims and acknowledge these limitations in the final version.